# Surface roughness during depositional growth and sublimation of ice crystals

Jens Voigtländer[3,*], Cedric Chou[1,2,*], Henner Bieligk[3], Tina Clauss[3,4], Susan Hartmann[3], Paul Herenz[3], Dennis Niedermeier[3], Georg Ritter[1], Frank Stratmann[3], and Zbigniew Ulanowski[1]

[1]School of Physics Astronomy and Mathematics, University of Hertfordshire, Hatfield, AL10 9AB, UK
[2]Department of Medicine, University of British Columbia, Vancouver, BC, Canada
[3]Leibniz Institute for Tropospheric Research, Permoserstr. 15, 04318 Leipzig, Germany
[4]DBFZ — Deutsches Biomasseforschungszentrum, Torgauer Str. 116, 04347 Leipzig, Germany
[*]These authors contributed equally to this work

**Correspondence:** Joseph Z. Ulanowski (z.ulanowski@herts.ac.uk)

**Abstract.** Ice surface properties can modify the scattering properties of atmospheric ice crystals and therefore affect the radiative properties of mixed-phase and cirrus clouds. The Ice Roughness Investigation System (IRIS) is a new laboratory setup designed to investigate the conditions at which roughness develops on single ice crystals, based on their size, morphology and growth conditions (relative humidity and temperature). Ice roughness is quantified through the analysis of speckle in 2-D light scattering patterns. Characterisation of the setup shows that a supersaturation of 20% with respect to ice and a temperature at the sample position as low as -40°C could be achieved within IRIS. Investigations of the influence of humidity show that higher supersaturations with respect to ice lead to enhanced roughness and irregularities of ice crystal surfaces. Moreover, relative humidity oscillations lead to gradual "ratcheting up" of roughness and irregularities, as the crystals undergo repeated growth-sublimation cycles. This memory effect also appears to result in reduced growth rates in later cycles. Thus, growth history, as well as supersaturation and temperature, influences ice crystal growth and properties, and future atmospheric models may benefit from its inclusion in the cloud evolution process, and allow more accurate representation of not just roughness but crystal size too, and possibly also electrification properties.

## 1 Introduction

Cloud properties and their effects remain the largest uncertainty in global climate models (Boucher et al., 2013). In particular, climate feedbacks of cirrus clouds, which cover 30% of the globe (Wylie et al., 2005) and 60-70% in the tropics (Sassen et al., 2009), are still not well understood. The main reasons of these uncertainties lie in the fact that ice crystals which compose cirrus clouds have widely varying morphology, size and concentration (McFarquhar and Heymsfield, 1997; Heymsfield et al., 2017) and therefore have different scattering properties, influencing in turn the radiative properties of the clouds (Hartmann et al., 1992; McFarquhar et al., 2000; Baran, 2012; Yang et al., 2015). Several cloud imaging probes like the Cloud Imaging Probe (CIP), Cloud Particle Imager (CPI) or various optical array probes (e.g. 2DC, 2DS) among others, have been developed over the last decades in order to characterise those parameters. However, these probes have difficulties characterising the size and shape

of smaller ice crystals, due to optical resolution limitations. Moreover, these probes do have a further limitation due to varying degree of shattering of ice particles on their inlets, only partly reduced by various mitigation measures (Ulanowski et al., 2004; Field et al., 2006; Connolly et al., 2007; Jensen et al., 2009; Korolev et al., 2011; Baumgardner et al., 2017). For these reasons, a new family of probes called collectively Small Ice Detector has been developed, which has open-path detection geometry

designed to reduce shattering, and relies on retrieving particle size and shape from scattering patterns instead of images (Kaye et al., 2008; Cotton et al., 2010; Ulanowski et al., 2014).

Further to the challenge of retrieving the size distribution and concentration of small ice crystals present in clouds, it has been shown on the basis of light-scattering models that irregularities of ice crystal surfaces, such as roughness and concavity can affect the light scattering properties (Yang et al., 2008b, a; Liu et al., 2013). Experiments performed on ice analogues have

shown that the asymmetry parameter of large ice analogue crystals possessing surface irregularities can be over 20% lower at visible wavelengths in comparison to smooth counterparts (Ulanowski et al., 2003, 2006). This in turn can result in large reductions in shortwave radiative forcing (Yi et al., 2013). Suggestions that atmospheric ice may depart from the idealized hexagonal crystal model have been made for several decades (Foot, 1988; Korolev et al., 1999; Auriol et al., 2001; Garrett et al., 2001) and there is now accumulating evidence that natural ice clouds can contain a significant proportion of, or can even

be dominated by, ice particles that substantially depart from the idealized smooth, hexagonal prism shape – for reviews and recent results see Ulanowski et al. (2014), Yang et al. (2015) and Hioki et al. (2016). Recent in-cloud measurements using the Small Ice Detector 3 (SID-3) during CONSTRAIN (Ulanowski et al., 2014) and MACPEX (Schmitt et al., 2016) have shown that most of the ice crystal encountered could be classified as rough. However, the conditions that lead to the development of crystal irregularities are still not well understood, because previous ice growth experiments tended to focus on growth rates and

crystal habits, rather than the fine detail that can contribute to light scattering properties.

To study the influence of growth conditions on ice crystal roughness, laboratory experiments simulating atmospherically relevant depositional growth and sublimation of ice crystals are required. In the present study, the newly developed experimental setup IRIS (Ice Roughness Investigation System) is introduced, which facilitates the exposure of a fixed ice crystal to different controlled relative humidities and temperature regimes. The system is used to carry out ice crystal growth experiments at

different supersaturation ratios, and to investigate the impact of repeated growth and sublimation cycles on surface roughness. The findings are discussed in the context of fundamental features of ice crystal growth and their atmospheric implications.

## 2   Methodology

### 2.1   Experimental setup

The experimental setup is a combination of a laminar flow tube and a laboratory version of the SID-3 instruments, which has been additionally equipped with an optical microscope. The laminar flow tube is used to precisely control the thermody-namic conditions in the optical measuring volume at the tube outlet. In the experiments, the ice crystals, generally single, are fixed within the measuring volume and exposed to thermodynamic conditions simulating single or multiple growth cycles at

various temperature and saturation ratio. A brief description of the setup, the operating principle and the thermodynamical characterisation is given in the following subsections.

### 2.1.1 Laminar flow tube

The laminar flow tube follows the principle of the Leipzig Aerosol Cloud Interaction Simulator (LACIS Stratmann et al. (2004); Hartmann et al. (2011)). It has been further developed to provide required conditions with respect to humidity and temperature at the outlet where the ice crystal is situated. The main part is a thermodynamically controlled laminar flow tube with a diameter of 15 mm and a length of 1.0 m. Both, the wall and the inlet temperature of the insulated flow tube can be precisely controlled by means of thermostats, which are operated in counter flow direction. The main differences between LACIS and IRIS are the extended mass flow range and the missing separated aerosol beam. The mass flow is adjusted by means of two mass flow controllers (Brooks 5850s, Brooks Instruments, Hatfield, PA, USA), controlling a dry (dew point between -60°C and -40°C) and a humidified gas flow in the range between 1 and 10 l/min standard temperature and pressure (STP), respectively. The humidification of the wet flow is done by water vapour transport through a micro-porous Nafion (sulfonated tetrafluoroethylene based fluoropolymer-copolymer) membrane. By operating the water cycle of the humidifier in counter flow direction, the air flow reaches a relative humidity (RH) of 100±0.03% at equilibrium. Both flows are combined in a small mixing bottle before entering the tube to ensure a well-mixed gas flow. The dew point temperature at the tube inlet then is defined by the mixing ratio and the dew points of both flows. A simplified schematic of the experimental setup is given in Fig. 1.

### 2.1.2 Optical system

The Leipzig Ice Scattering Apparatus (LISA) is a laboratory version of the Small Ice Detector 3 (SID-3, also known in its other laboratory version as the Particle Phase Discriminator - PPD), which allows the differentiation of ice crystals from water droplets, as well as ice crystal size and shape characterisation (Kaye et al., 2008). In addition, it also allows the characterisation of ice crystal irregularities on the basis of the two-dimensional (2-D) speckle pattern distribution (Ulanowski et al., 2014). 2-D scattering patterns are collected by LISA via an intensified CCD camera at scattering angles from 6 to 25° in an annular shape, which ensures that the bright feature associated with the familiar halo occurring for ice prisms at the scattering angle of 22° is included. The smaller angles in the central area are not captured due to the presence of a beam stop. The camera images are digitized as 12-bit TIFF files allowing a wider dynamic range with a fixed intensifier gain value, as opposed to the 8-bit JPEG files used during CONSTRAIN (Ulanowski et al., 2014). In addition, a CMOS camera (Prosilica GC1280, Allied Vision Technologies, USA) attached to a 20-times magnification long working-distance microscope objective (20x Mitutoyo Plan Apo) via an Infinitube right-angle adaptor with fibre-optics in-line illumination (Infinity Photo-Optical Company, USA) was added to LISA in order to visualise the evolution of the ice crystal. Ice crystals were monitored by the camera at 27 frames per second, ensuring no loss of information during fast humidity cycles. Fig. 2 shows a simplified schematic of LISA.

## 2.2 Numerical simulations and thermodynamic characterisation

The thermodynamic conditions at the tube outlet were extensively studied by means of computational fluid dynamics (CFD) simulations of the laminar flow tube, and by measurements of flow velocity, temperature and dew point at the tube outlet. Both the numerical simulations and the measurements have been done to characterise the experimental setup, as well as to demonstrate the fast control of temperature and supersaturation in the measuring volume.

The numerical simulations were done with the commercially available CFD code Fluent (Ansys Inc., USA). The Fluent model is a general purpose FVM (finite volume method) CFD model allowing the simulation of a wide range of small scale fluid flow problems. Here, the flow through the flow tube was simulated including the coupled processes of mass and heat transfer. With respect to the geometry and the laminar flow regime, the simulations were done on a 2-dimensional axisymmetric Cartesian grid by means of a pressure based steady state solver. Additional information about the numerical model, which has already been successfully applied to the characterisation of the laminar flow tube LACIS, can be found for example in Stratmann et al. (2004); Voigtländer et al. (2007) and Hartmann et al. (2011).

To illustrate the operating principle of the laminar flow diffusion channel, calculated thermodynamic profiles along the tube axis are shown in Fig. 3. Generally, the thermodynamic conditions in the measuring volume at the flow tube outlet depend on mass and heat transfer to the tube wall. Since the tube wall temperature is adjusted to lower values than the inlet temperature the gas flow temperature along the tube axis decreases due to heat conduction (Fig. 3, top right). Depending on both, the temperature gradient and the inlet dew point, supersaturation with respect to water and/or ice can be achieved (Fig. 3, bottom). If the residence time of the gas flow (controlled by the mass flow rate) is large enough, the gas flow cools down until thermodynamic equilibrium with the tube wall is reached. Conversely, for a sufficiently fast flow equilibrium will not be reached. Consequently, the thermodynamic conditions at the tube outlet are determined by the total mass flow, the wall and inlet temperatures, as well as the inlet saturation ratio. Fig. 4 shows that for a total flow rate between 4 and 10 l/min, an inlet temperature of 20°C and a wall temperature of -30°C, the temperature at the tube centre outlet increases almost linearly by about 10°C with increasing flow. As a consequence the temperature in the measuring volume can be varied by at least 10°C on a very short time scale (about 2 s) by varying the total flow rate. In contrast, the temperature values for a flow rate smaller than about 3 l/min indicate that the residence time becomes sufficiently long to approach the thermodynamic equilibrium state. In this case, the thermodynamic conditions can no longer be controlled by varying the flow rate. Since control of the conditions by adjusting the thermostats (wall and inlet temperatures) is much slower, a flow rate of about 3-4 l/min represents the lower practical limit for the experiments. Because temperature is usually kept constant in typical experiments, the saturation ratio is controlled by varying the inlet dew point. As mentioned before, this can be also done on a short time scale (about 5 s) by controlling the ratio of the dry and the wet air flow while the total flow is kept constant.

The thermodynamic conditions at the tube outlet have been extensively characterised using measurements of the flow velocity, temperature and dew point. The flow characterisation was done applying hot wire anemometry (Dantec Dynamics A/S, Skovlunde, Denmark). By means of a miniature, single-axis probe, the velocity magnitude was determined at several points along a cross sectional profile of the tube outlet showing a parabolic laminar velocity profile without any back flows. In the

measuring volume, the flow velocity increases linearly from approx. 0.5 to 2.0 m/s for total flow rates between 3 and 10 l/min. For typical experimental conditions with flow rates around 5 l/min the resulting flow velocity is about 1 m/s. Measured and calculated flow velocities were found to be very similar (see supplement material).

Temperature measurements were done at the outlet of the laminar flow chamber using calibrated (to an accuracy of ±0.01 K.) external resistance thermometers Pt100, as well as K-type thermocouple sensors. Fig. 4 shows measured temperatures as a function of the total volume flow at selected wall temperatures (-20°C, -30°C and -40°C) in comparison to Fluent simulation data for an inlet temperature of 20°C. It can also be seen that both data sets are in a good agreement. Additionally, an extended data set of temperature measurements is shown in the supplement material.

The dew point temperature (respectively frost point temperature and relative humidity) was characterised using a dew point mirror (model Dew Point Mirror 973, MBW Calibration, Wettingen, Switzerland). An example of measured relative humidity with respect to ice (RHi) is shown in Fig. 5. Therein, the RHi is depicted in dependence of the wet flow. In this example, which represents typical experimental conditions, the wall temperature was at -30°C, the inlet temperature at 20°C, and the inlet dew point temperature at 19.5°C. The total flow was kept at a constant value of 5 l/min. According to Fig. 4, the resulting temperature was about -27.5°C. The saturation ratio rises steeply with increasing fraction of the humidified flow. Fig. 5 demonstrates that the saturation ratio in the measuring volume can be varied between sub- and supersaturated conditions (with respect to both water and ice). Here, the saturation ratio with respect to ice ranges from 0.75 (0.1 l/min wet flow) up to 1.2 (2 l/min wet flow).

In the experiments it was also found that for low wall temperatures and supersaturated conditions (with respect to ice) the conditions in the sampling volume changed slowly with time. Thereby, the lower the wall temperature and the higher the supersaturation, the faster the conditions in the tube change. Consequently, experiments at very high saturation ratios (>1.20 wrt. ice) were avoided . Due to ice formation at the tube wall, the observed temperature and the dew point increased at the tube outlet. This is most likely caused by the growing ice shell at the tube wall, which acts as a thermal insulator suppressing the temperature gradient and hence the diffusional processes in the flow tube. For example, considering a wall temperature of -40°C, a total flow of 7 l/min and a wet flow of 0.9 l/min, resulting in a saturation ratio of about 1.2 with respect to ice in the beginning, the temperature increases from about -31.7°C to -31.0°C and the frost point from about -29.5°C to -29.0°C within 20 min. Fluctuations of both the temperature and the dew/frost point have an impact on the saturation ratio and therefore on the growth rate of the observed ice crystal, the ice layer at the tube wall can also act as an extra source for water vapour, if the conditions are changed from super- to subsaturated conditions. It has to be concluded that for supersaturated experiments, a detailed quantitative characterisation of the saturation ratio on the basis of dew point measurements is not possible and that additional methods are needed to evaluate the saturation ratio during the experiments. This is done by observation of the ice crystal with the optical microscope. Since the ice crystal growth process is highly sensitive to the prevailing thermodynamic conditions, i.e. the saturation ratio determines the ice crystal growth rate, the mass flow controllers can be adjusted according to the microscope images. In this way the settings corresponding to the point of equilibrium between the crystal and the vapour can be found, to act as a reference point.

## 2.3 Sample preparation

Laboratory investigation of ice crystal surface properties requires the formation of the initial ice crystal from a foreign particle
termed an ice nucleating particle (INP). For experiments done at IRIS, a single INP is attached to the tip of a thin glass fibre.
These very thin tips of about $2\,\mu$m in diameter were pulled from 1 mm diameter borosilicate rods using a micropipette glass
puller. The glass was cleaned and hydrophobically coated following the procedure described by Dymarska et al. (2006). The
glass fibre was then attached to a micromanipulator (Singer Mk.1, Singer Instruments, UK) which was used to pick the INP
(usually of several micrometres in size) that had been deposited on a microscope glass slide under a microscope (Zeiss Primo
Vert, Germany).

## 2.4 Scattering pattern analysis

The 2-D scattering patterns are characterised by quantifying their brightness distributions and texture (Ulanowski et al., 2014).
In brief, image texture can be retrieved by using the Grey-Level Co-occurrence Matrix (GLCM) which consists of pairing each
grey-level pixel with the nearest neighbour pixels in four directions (Haralick et al., 1973). GLCM has been used in the past
to assess surface roughness based on laser speckle images (Lu et al., 2006). It was found and discussed in Ulanowski et al.
(2014) that among the four features of the GLCM (contrast, correlation, homogeneity and energy, also known as uniformity or
angular second moment), energy is the parameter which has the strongest correlation to roughness but is also less biased by
external factors (e.g. image brightness change due to camera gain change). Statistical measures describing image brightness
distribution are obtained by two methods, firstly by calculating the ratio of root-mean-squared brightness to its standard de-
viation (RMS / SD) (Jolic et al., 1994) and secondly by calculating the kurtosis of the brightness distribution. This leads to a
"combined roughness" measure, following the expression:

$$0.7 - \frac{2E}{3} - \frac{(logK)}{6} + \frac{\frac{RMS}{SD}}{4000}, \tag{1}$$

where $E$ is the energy derived from the GLCM and $K$ the kurtosis. The combined measure is dependent on the number
of independent "scattering centres" present on the surface of the crystal, so it reflects the overall complexity of ice crystals,
including both small-scale roughness as well as larger-scale structure, such as that found in so-called "polycrystals" (Ulanowski
et al., 2012, 2014). The combined roughness has been tested on ice analogues and mineral dust with various surfaces and results
that are reported in Ulanowski et al. (2014) show that it does provide a good estimation of a particle surface irregularity. In
addition, the size of the ice particles, which is inversely proportional to the average area of speckle spots, is retrieved and is
used throughout the present work to determine crystal size. The size measured in this way represents the diameter of equal area
circle projected along the line parallel to the laser beam (Ulanowski et al., 2012).

Time-lapse videos showing the evolution of LISA 2-D scattering patterns during cyclic ice growth experiments are shown in
the supplementary material (S1 and S2). Broadly, the size of the speckle spots visible in the patterns is a reflection of ice crystal
size (strictly speaking its inverse), and the amount of speckle represents crystal roughness. In the videos, growth periods are

characterized by the spots shrinking and generally moving inwards, and the opposite occurs during sublimation periods. The presence of isolated bright spots or bands is an indication of flat crystal facets, while spots covering a large proportion of the pattern signify the presence of roughness or high complexity.

## 3    Results and discussion

In the following, experiments are presented and discussed first, addressing two aspects: the influence of supersaturation, and of
regrowth cycles on the ice crystal surface roughness measures. However, we note that in general other factors may also influence crystal morphology, like the type, shape and size of the INP, the mode of ice nucleation (homogeneous, immersion freezing or deposition nucleation), temperature and ventilation (as represented by the fall speed). For example, the AIDA experiments mentioned above also indicate that homogeneous nucleation can lead to crystals with strongly rough surfaces (Schnaiter et al., 2016). Furthermore, it has been shown that droplets freezing at lower temperatures are more likely to grow into complex
"polycrystals" (Pitter and Pruppacher, 1973; Bacon et al., 2003). Since homogeneous nucleation occurs, by definition, at low temperatures it may lead to the formation of more imperfect crystals; however, we must note that the temperature effect may be secondary to the influence of the high saturation ratios that are necessary to initiate homogeneous nucleation. Investigation of the influence of these parameters is beyond the scope of this paper, but might be addressed in future work.

Fig. 6 shows examples of 2-D scattering patterns from smooth and rough ice columns to illustrate what patterns could be
classified as originating from smooth or rough ice crystals in the following discussion. It can be seen that the main characteristic of a rough particle/crystal is the speckle captured by the CCD camera, which is the outcome of complex interference between scattered waves originating from multiple regions of the particle (Ulanowski et al., 2012). Smooth crystals on the other hand do not produce strong speckle, as scattering is dominated by fewer, distinct interactions with the particle, which tend not to produce the interference giving rise to speckle patterns; these interactions are more akin to the reflections and refractions of
classical geometric optics but enhanced by diffraction, which leads to the appearance of arc-like features in the 2-D patterns (Clarke et al., 2006). More examples obtained from ice analogues and other types of smooth and rough or irregular particles can be seen in Ulanowski et al. (2014).

Most of the noise visible in the retrieved crystal size and roughness measure in the experiment time series has been found to originate from the gas flow of the laminar flow tube which creates vibrations of the fibre. Slight vibrations of the sample
result in scattered data points. However, in all the cases, the trend is discernible and becomes clearer after applying locally weighted scatterplot smoothing (LOESS, Cleveland and Devlin (1988)) to the data points in order to obtain a trend curve for each experiment.

### 3.1    Slow and fast growth

Fast crystal growth tends to lead to the emergence of roughness on crystal surfaces. The conventional view of ice (and other material) growth suggests that regular, smooth crystals grow at low supersaturation, where the growth rate is slow enough for the deposited molecules to diffuse laterally on facets to well-separated attachment sites at steps, kinks, and ledges. In contrast,

fast growth promotes attachment anywhere on crystal surface through 2-D nucleation, and potentially also step bunching, re-
sulting in roughness (Mason et al., 1963; Beckmann, 1982; Dash and Wettlaufer, 2003; Pantaraks and Flood, 2005; Dash et al.,
2006; Ferreira et al., 2008; Flood, 2010; Sazaki et al., 2010). Fast growth can moreover lead to the creation of defects and ion-
ization, which further promote irregular growth (Dash et al., 2001; Dash and Wettlaufer, 2003). Furthermore, the mechanisms
leading to surface roughness are likely to depend on growth temperature, in the same way that the gross crystal habit does due
to different growth rates on the basal and prismatic facets (Mason et al., 1963; Bailey and Hallett, 2004). Hence roughness
may arise on different facets at different temperatures. Also temperature dependent is the role in ice growth of the quasi-liquid
layer (QLL). Its thickness, amount of disorder and hence importance diminish with decreasing temperature, with some studies
indicating little impact at the temperatures used here but, as yet, there is much disagreement between molecular dynamics
modelling and measurements of QLLs (Dash et al., 2006; Gladich et al., 2015; Michaelides et al., 2017). Supersaturation was
identified as one of the main parameters controlling the surface roughness in experiments conducted for ice by Hallett (1987),
and complex polycrystals tend to dominate ice habits at high supersaturations (Bacon et al., 2003; Bailey and Hallett, 2004).
There is evidence from recent, dedicated experiments in the AIDA cloud chamber that increasing the maximum supersatura-
tion achieved during chamber expansions leads to increased roughness, as indicated by SID-3 measurements (Schnaiter et al.,
2016).

Since the supersaturation controls the growth rate but could not be determined directly with high accuracy in our experiments,
several slow and fast ice crystal growths experiments at -40°C were performed. Thereby, slow ice crystal growth could be
observed at low, and faster growth at higher supersaturation. For comparability, the shape of the investigated ice crystals was
kept similar, hence very high supersaturation ratios resulting in the formation of complex ice crystals were excluded, and only
single columns were considered. This means that even in the fast growth experiments the supersaturation wrt. ice was less than
about 20% (typically between 10 and 20%). The slow growth experiments were done closer to saturated conditions (typically
about 5% supersaturation wrt. ice).

Fig. 7 shows the growth of two regrown ice crystals exposed to different level of supersaturation at -35°C. Based on the flow
rate ratio, which was varied between 0.7/4.3 l/min (wet/dry, slow) and 1.0/4.0 l/min (fast), the difference in relative humidity
wrt. ice was about 10 percent (compare Fig 5). The crystals can be compared directly as they grow from 20 $\mu$m to 29 $\mu$m,
after fitting trend curves using LOESS. The crystal regrown at a higher saturation ratio shows a greater and steeper roughness
increase – from 0.27 to 0.39 over 23 seconds, an increase of 0.12, as compared to the crystal exposed to lower humidity – from
0.29 to 0.37 over 100 seconds, an increase of 0.08. In Fig. 8 the relationship between the temporal rate of change of roughness
and the growth rate is depicted for ice crystals in the size range from 20 $\mu$m to 80 $\mu$m. Fifteen different growth experiments
leading to the formation of simple columns were performed to investigate this relationship with twelve cases where the initial
crystal was sublimated and regrown. The correlation was found to be strong, with the coefficient of determination $R^2$ of 0.82,
implying that higher degree of supersaturation leads to faster roughness evolution, i.e. quicker formation of more irregular ice
surfaces.

Concerning possible mechanisms for the emergence of roughness during crystal growth, while the general ones discussed
above are likely to play a role, an additional one may be important specifically for water ice. Laboratory experiments indicate

that ice at atmospherically relevant temperatures, rather than being formed purely from the hexagonal crystallographic phase,

can contain numerous stacking faults, where adjacent molecular layers are stacked in the cubic instead of hexagonal sequence, leading to "stacking-disordered ice". Such structure is associated with the lower symmetry, trigonal crystallographic space group P3m1 (Hansen et al., 2008; Murray et al., 2015) and can lead to the production of scalene ice crystals (Kuhs et al., 2012; Murray et al., 2015). Stacking disorder has been associated with the presence of macroscopic kinks and roughness on prismatic facets (Kuhs et al., 2012). It can occur even at both cold and warmer temperatures (Malkin et al., 2012, 2015). While

stacking-disordered ice tends to anneal to hexagonal ice at higher temperatures (Kuhs et al., 2012), its presence in the early stages of growth may nevertheless influence crystal shape, even if the phase is absent from "mature", crystals. However, we must note that on the basis of molecular modelling the stacking-disordered phase may be less likely to form during growth from vapour than during freezing (Hudait and Molinero, 2016).

## 3.2 Roughness due to humidity cycles

Another process that could influence the roughness of ice crystals is the exposure to several depositional growth and sublimation cycles, which can occur in the atmosphere (Nelson, 1998; Korolev et al., 1999; Ulanowski et al., 2014). Fig. 9 shows an example of how such repeated cycles performed at -30°C can lead to roughness in an initially smooth column. After the initial growth, the saturation ratio with respect to ice was between about 1.0 and 1.05 (based on flow rate ratio, which was set to values between 0.8 l/min and 1.0 l/min wet flow, and 4.2 l/min and 4.0 l/min dry flow, see Fig. 5) and was decreased in order to partly

sublimate the crystal. At $t = 140$ s, it can be seen that the column has shrunk and shows a relatively smooth surface with a combined roughness value of about 0.3. The corresponding 2-D scattering pattern is also typical of a relatively smooth column, as can be seen by comparison with Fig. 6. Upon re-growing the crystal, irregularities started forming on the surface, as can be observed in the microscopy image and in the increase of the combined roughness at $t = 250$ s. The ice crystal then was shrunk and regrown again, ending with a combined roughness of about 0.6. A video showing the evolution of the LISA 2-D scattering

patterns during the experiment in Fig. 9 is shown in the supplementary material (S1). The 10 minute period is condensed into one minute, but the time displayed corresponds to the abscissa in Fig. 9. In the video, the inward and outward drifting of speckle spots corresponds to crystal growth and sublimation, respectively.

A second example is shown in Fig. 10. In this experiment an illite particle on the tip of the glass fibre nucleated an ice crystal at a lower temperature of -40°C. The saturation ratio with respect to ice was similar and between about 1.0 and 1.1 (after the

initial growth, values based on flow rate ratio, which was set to values between 0.6 l/min and 0.8 l/min wet flow, and 4.4 l/min and 4.2 l/min dry flow, see Fig. 5). A video showing the evolution of the 2-D patterns during the experiment in Fig. 10 is shown in the supplementary material (S2); as in the previous video the time is speeded up by a factor of 10. As in Fig. 9, rougher features appear during the second growth cycle, but then also in the additional third cycle. Although more experiments would be required to get robust statistics, these observations indicate that the more growth-sublimation cycles are performed, the rougher the crystal can become. The increase observed in the final roughness appeared to be the outcome of an asymmetric, irreversible character of the cyclic growth process, i.e. the increase in roughness during a growth phase was not mirrored by

an equal decrease of roughness during the subsequent sublimation phase. Thus, the overall outcome was a gradual "ratcheting up" of roughness.

We note that similar behaviour has been postulated by Nelson (1998) who stated that due to asymmetry between the growth and sublimation processes, primarily because of different energy barriers to step formation – weaker in the case of sublimation – repeated growth-sublimation cycles would lead to progressively more complex ice crystals. Furthermore, in experiments carried out under atmospherically-relevant air pressures, crystals having undergone more than one growth cycle tended to develop more faults (Beckmann, 1982). Korolev et al. (1999) stated that cycling of growth and sublimation caused by mixing and small scale vertical motions might be a possible route for the formation of irregular crystals. This conjecture is supported by our experiments. It was also suggested by the same authors that "sublimation may cause numerous irregularities on the surface of the ice crystal". However, while sublimation can in some cases lead to increased roughness, as demonstrated by experiments carried out in Scanning Electron Microscopy (SEM) chambers (Cross, 1969; Pfalzgraff et al., 2010; Neshyba et al., 2013; Ulanowski et al., 2014; Magee et al., 2014), the cyclic growth described here tended to result in a reduction in roughness during each sublimation phase.

The apparent disparity between our observations and SEM experiments can be accounted for by the fact that growth in the absence of air that takes place in a SEM chamber, instead of being limited by vapour diffusion as is the case for ice at tropospheric pressures, becomes limited by the attachment kinetics. This distinction is known to lead to different growth rates as well as habits (Beckmann, 1982; Kuroda and Gonda, 1984; Libbrecht, 2017). Consequently, during SEM observations water molecule removal can take place anywhere on facet surfaces, leading to pronounced roughness. Moreover, the bunching of elementary molecular steps, possibly due to the Schwoebel effect (Misbah et al., 2010), can result in the creation of larger, microscopic (as opposed to elementary) steps that can be seen in SEM micrographs (Cross, 1969). In contrast, in the diffusion-limited regime sublimation tends to remove material near crystal edges and vertices, which can lead to reduced roughness (and rounding), as was the case in our observations. However, for very small crystals the relative impact of diffusion is diminished anyway (Yokoyama and Kuroda, 1990), so we conjecture that sublimation of such crystals might potentially lead to increased roughness even at tropospheric pressures.

A further difference between the diffusion limited and kinetics-limited growth is that the former can lead to increased numbers of faults (Beckmann, 1982) and dendritic, skeletal, or needle-shaped crystals, while the latter tends to produce more perfect, smooth, isometric prisms (Gonda, 1976, 1977) – with similarity to the SEM chamber experiments. Thus stronger departure from smooth, regular crystal shapes – roughness in a general sense – can be expected at tropospheric pressures, provided that the crystals are large enough compared to the mean free path of the water molecule at the given pressure (Yokoyama and Kuroda, 1990).

Another feature of the growth process may be borne out by the cyclic growth experiments. Careful examination of the retrieved crystal size shown in Fig 9 indicates markedly slower growth in later cycles, despite similar supersaturation levels. A similar effect was in the past observed for crystallization from solution - a reduction of the growth rate of sucrose crystals that previously experienced a period of rapid growth (Pantaraks and Flood, 2005; Ferreira et al., 2008; Flood, 2010). Thus a "memory effect" may be present, which not only leads to increased roughness in subsequent growth cycles but results

in reduced growth rates under identical supersaturations. This reduction can at first sight be contrary to expectations: high roughness, reflected by high density of surface defects, might lead to increased growth rate under kinetically-limited growth conditions. However, this would not occur when growth is diffusion limited, as is the case at troposperically relevant air pressures. On the contrary, defects resulting from fast growth may inhibit the incorporation of molecules on the crystal surface, possibly through the introduction of impurities (Ferreira et al., 2008; Flood, 2010). Nevertheless we caution that this potentially
very important finding must be confirmed through carefully controlled experiments, to eliminate the possibility that the reduced growth rate may have been caused by instrumental factors, such as a reduction of supersaturation in later cycles.

## 3.3   Atmospheric implications

It is important to mention that certain atmospheric parameters cannot be reproduced in the current experimental setup. These parameters, like air pressure (Neshyba et al., 2013), radiative heating/cooling or fall velocity can potentially influence the
shape of the crystals and their complexity (Hallett, 1987). For example, ice crystal growth is significantly influenced by the ventilation effect (Westbrook and Heymsfield, 2011); high velocities favour plate-like over dendritic habits (Keller and Hallett, 1982); roughness can emerge at lower supersaturations for falling crystals than for stationary ones (Yokoyama and Kuroda, 1990). As stated above, the flow velocity in IRIS is about 1 m/s, which is slightly higher than the typical fall velocity of ice crystals in the atmosphere; consequently, at more atmospherically relevant velocities, crystal roughness might conceivably
become greater than observed here. Airflow could also create asymmetry of the ice crystal shape (Takahashi and Mori, 2006) and therefore the same crystal can have quite different properties depending on its orientation.

Nevertheless, the combined roughness values obtained in the experiments shown here during the two re-growth cycles (values between 0.55 and 0.65) are comparable to the values encountered in mid-latitude cirrus and mixed-phase clouds during CONSTRAIN (Ulanowski et al., 2014), indicating that the results are at least qualitatively comparable.
Very high supersaturation regions can be encountered in clean air at cirrus altitudes or within cirrus clouds. Several studies reported supersaturations with respect to ice up to 140% (the onset of homogeneous ice nucleation) within a temperature range from -33°C to -73°C (Gierens et al., 1999, 2000; Ovarlez et al., 2002; Spichtinger et al., 2003, 2004; Krämer et al., 2009). It is very likely, as shown in our experiments, that at higher supersaturation rougher crystals tend to develop instead of smoother ones. The low temperatures at which homogeneous nucleation becomes dominant may be an additional factor
leading to increased crystal roughness in the broad sense, as droplets freezing at lower temperatures are more likely to grow into complex "polycrystals" (Pitter and Pruppacher, 1973; Bacon et al., 2003). On the other hand, high number concentrations of homogeneously nucleated ice lead to quick depletion of available water vapour, hence the ice growth rate can be expected to rapidly peak and then decline. We suggest that disentangling these conflicting influences may be possible through cloud chamber studies.

Humidity variation is omnipresent in cirrus. For example, Kübbeler et al. (2011) and Krämer et al. (2009) show that saturation with respect to ice can depart widely from equilibrium. Therefore, mixing and small scale vertical motions in clouds, leading to the presence of many depositional growth and sublimation regions, may be a possible mechanism by which roughness emerges
(Korolev et al., 1999). In both cirrus and mixed phase clouds, high-resolution modelling predicts that ice particle trajectories can

contain multiple super- and subsaturated regions (Flossmann and Wobrock, 2010; Kübbeler et al., 2011). Such inhomogeneity can be seen as arising from temperature fluctuations due to turbulence at various scales or gravity waves. Indeed, agreement between models and cirrus ice measurements is improved by the inclusion of such fluctuations, in comparisons across a broad range of geographical latitudes (Hoyle et al., 2005; Engel et al., 2013; Jensen et al., 2013). Provided that the ice crystals do not fully sublimate following such cyclic growth, this process would lead to creating rougher ice crystals. Moreover, longer-lived (older) clouds may contain higher incidence and/or intensity of roughness than short-lived ones. The history of individual ice crystals may also impact their subsequent growth rates, as discussed in section 3.2. Therefore future microphysical or parameterized cloud models may be improved by the introduction of such a memory effect, in addition to temperature and humidity.

Finally, we note that rough ice surfaces are associated with stronger electrical charging (Caranti and Illingworth, 1983; Dash et al., 2001; Dash and Wettlaufer, 2003), hence the presence of roughness may influence storm electrification. If so, measures of roughness based on 2-D scattering patterns could be an indicator of the cloud electrification potential of ice crystals, since they are sensitive to the presence of multiple scattering centres on ice surfaces, corresponding to the unevenness of the surfaces.

## 4   Conclusions and outlook

An experimental system was developed to investigate light scattering properties of single ice crystals grown on a glass fibre in dependence on the prevailing thermodynamic conditions. The system, called Ice Roughness Investigation System (IRIS), is based on the laminar flow tube LACIS and a laboratory version of the Small Ice Detector 3 (SID-3), which was additionally equipped with an optical microscope. The thermodynamic conditions during the experiments are controlled by varying the dry and wet gas flows passing through the flow tube, allowing changes of temperature and saturation ratio over a wide range on a time scale of less than 5 s.

First investigations of the impact of the saturation ratio and cyclic growth on the development of ice crystal roughness show that repeated depositional growth-sublimation cycles lead to progressively increasing ice crystal roughness, as indicated by 2-D scattering patterns. Results also show that the higher the supersaturation, the faster the increase in roughness. Moreover, crystal growth rate appears to be lower later in the cycles, once roughness has developed, hinting at an additional memory effect.

Further experiments should be performed to achieve a better understanding of the influence of supersaturation on the development of rough ice crystals at different temperatures. In addition, it is also of high importance to investigate what is the impact of the nature of the ice nucleating particle and the nucleation mechanism on the development of roughness. In particular, cloud chamber experiments should elucidate the influence of the dynamics of the homogeneous nucleation process, whereby large initial supersaturation leading potentially to increased roughness (due to fast growth) may be counteracted by reduced supersaturation due to the presence of high ice crystal concentrations (i.e. slow growth). Experiments comparing the degree of roughness from repeated humidity cycles with different frequencies (i.e. the same growth and sublimation rates (i.e. humidity amplitude) and total duration but shorter or longer cycles) would elucidate whether the lifetime of the cloud or the frequency of

fluctuations is more important for roughness development. Finally, the sensitivity of ice crystal growth rate to growth history, in particular the resulting roughness, should be investigated, with a view to improving future cloud models. Such improvements

may allow more accurate representation of not just roughness but crystal size too, and hence characteristics such as radiative properties, cloud lifetime, precipitation development, and possibly also cloud electrification properties.

*Author contributions:* C.C, J.V., P.H., H.B. designed and carried out the experiments including sample preparation and data analysis with assistance from T.C. J.V. performed the computational fluid dynamics simulations. Z.U. conceived and supervised the project and provided crystal property measurement techniques and interpretation of ice growth processes. F.S., Z.U, J.V.

developed the main conceptual ideas and the technical details of the experiments and the experimental setup IRIS. C.C., J.V., Z.U. wrote the manuscript with support from P.H., H.B., F.S, T.C., D.N., S.H. and G.R. All authors discussed the results.

*Competing interests:* The authors declare that they have no conflict of interest.

*Acknowledgements:* This work was supported by the UK Natural Environment Research Council grant NE/I020067/1 (ACID-PRUF) and the EU Eurochamp-2 scheme grant E2-2011-12-06-0065. The concept of LISA was proposed by Alexei

Kiselev, and the instrument itself was designed and built by Edwin Hirst at the University of Hertfordshire. We also acknowledge Christabel Tan for preparing the chemical solution used to clean the fibres.

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

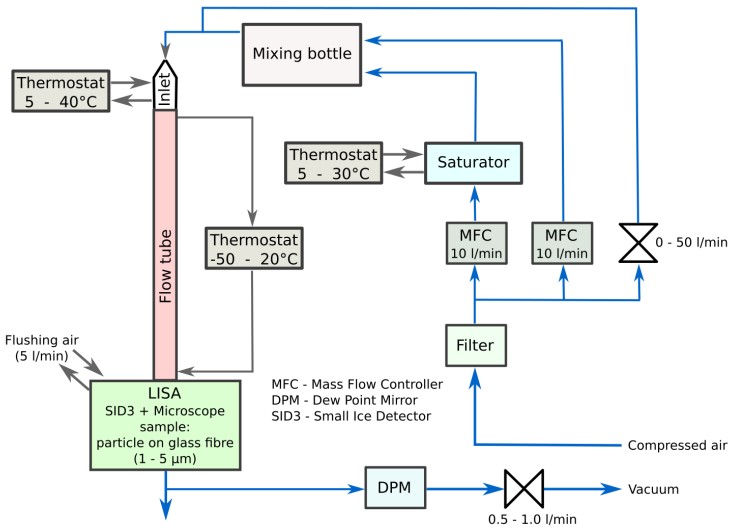

**Figure 1.** Simplified schematic of the Ice Roughness Investigation System (IRIS).

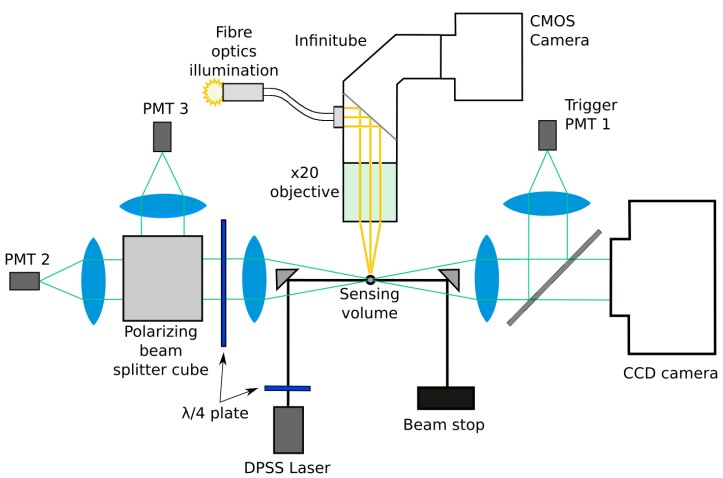

**Figure 2.** Schematic of the Leipzig Ice Scattering Apparatus (LISA), the optical system of IRIS.

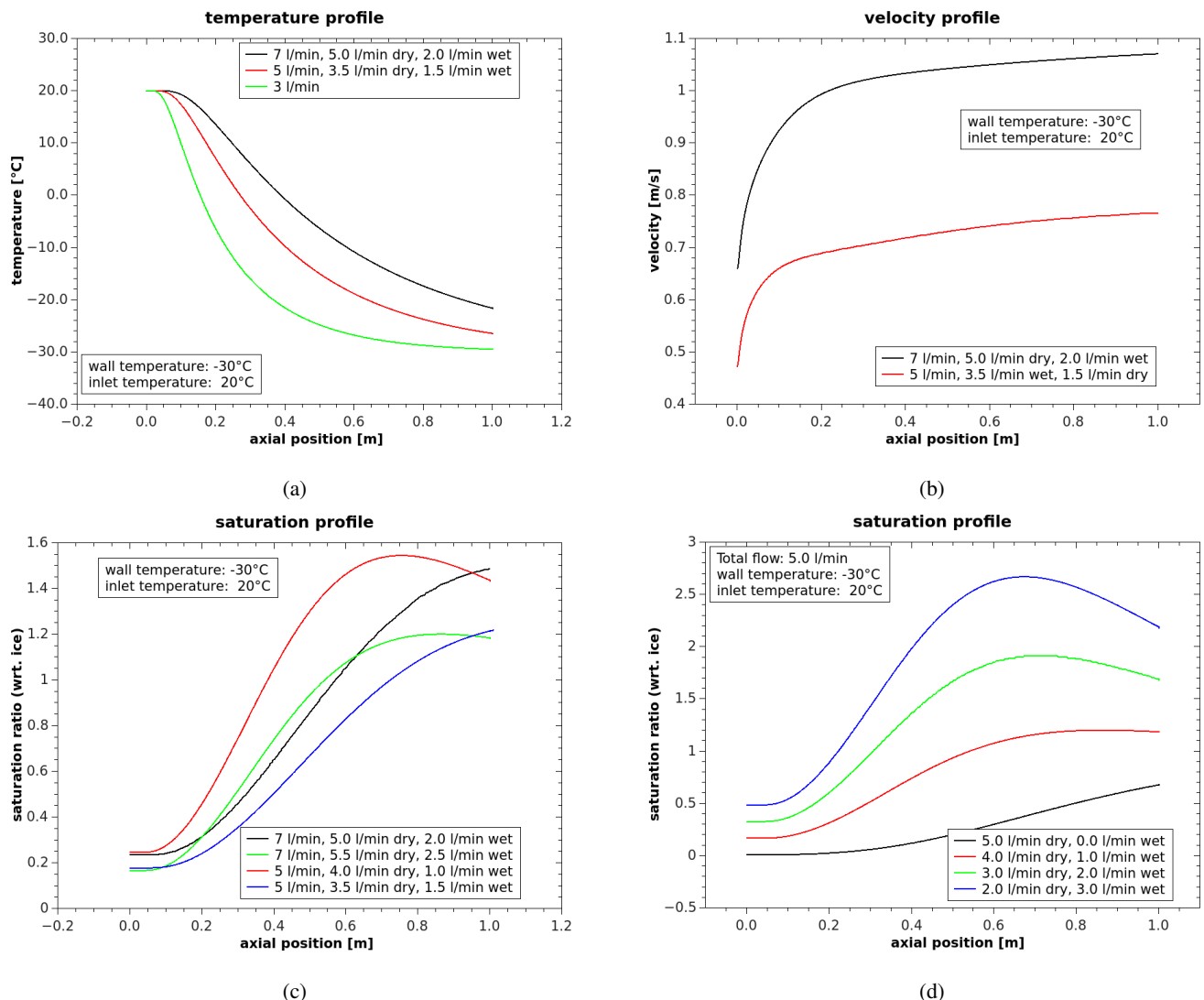

**Figure 3.** Examples of calculated flow velocity (a), temperature (b) and saturation ratio (bottom) profiles along the tube axis. The simulations were done with the CFD code Fluent. The tube length in the simulations was 1.2m, while it is only 1.0m in the experimental setup.

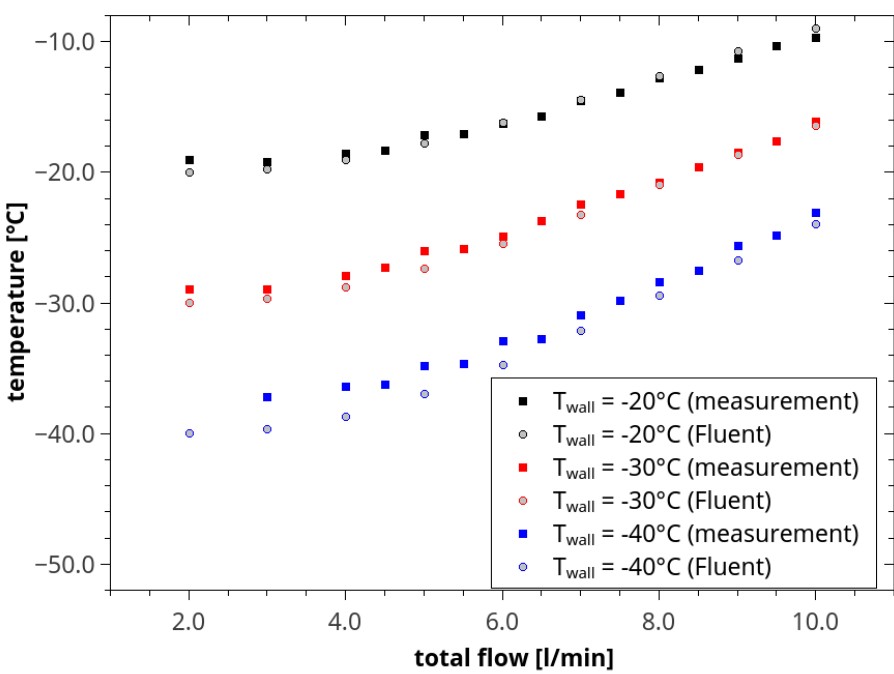

**Figure 4.** Comparison between measured and calculated temperature in dependence of the total flow rate using three different wall temperatures. The solid lines represent interpolation of the experimental data.

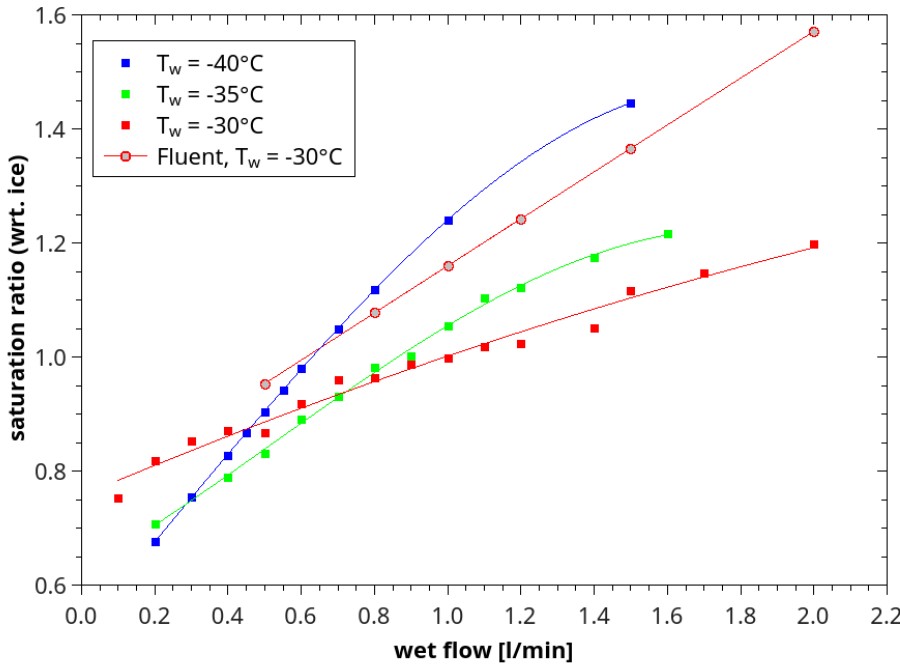

**Figure 5.** Measured (circles) and modelled (squares) relative humidity (wrt. ice) in the sampling volume of LISA. Measured temperatures in the observation volume of LISA. The total flow rate was 5 l/min. The values are given in dependence of the wet flow rate.

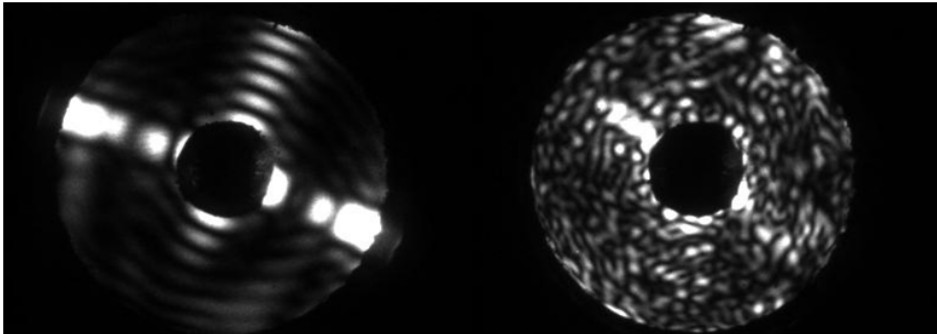

**Figure 6.** 2-D scattering patterns of a smooth column (left panel) and a rough column (right panel). The patterns were produced using the SID3 instrument in the AIDA cloud chamber during growth at low (left) and high (right) supersaturation (Schnaiter et al., 2016).

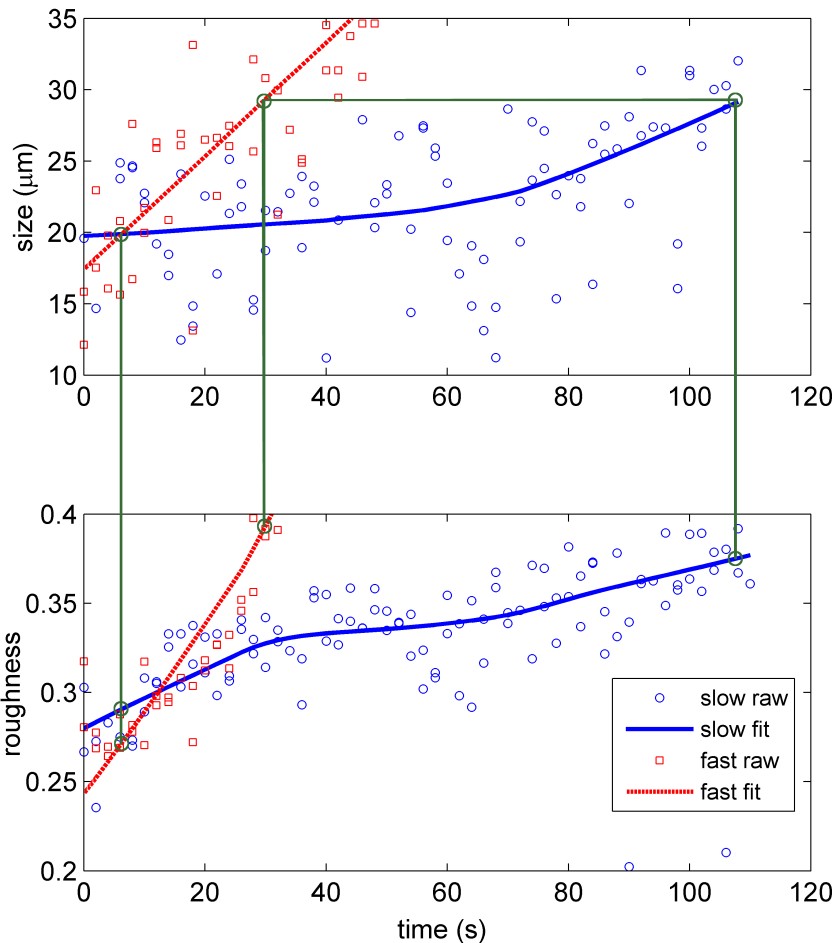

**Figure 7.** Size and roughness of two pre-existing ice crystals exposed to different supersaturations (difference about 10 percent): slow growth (blue symbols and lines) and fast growth (red symbol and lines); the trend curves were fitted using LOESS. The crystals can be compared directly as they grow from 20 $\mu$m to 29 $\mu$m. Based on the flow rate ratio, which was varied between 0.7/4.3 l/min (wet/dry, slow) and 1.0/4.0 l/min (fast), the difference in relative humidity wrt. ice was about 10 percent (compare Fig 5).

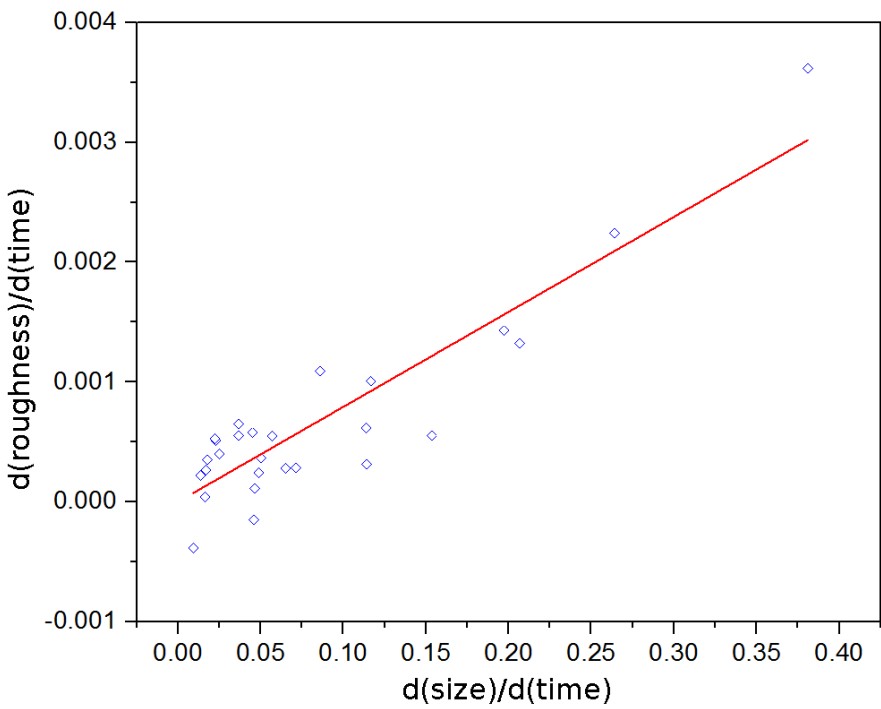

**Figure 8.** Rate of increase of roughness as a function of the growth rate (indicating the degree of supersaturation in the flow tube). Each point corresponds to a separate growth experiment. The coefficient of determination $R^2$ is 0.82.

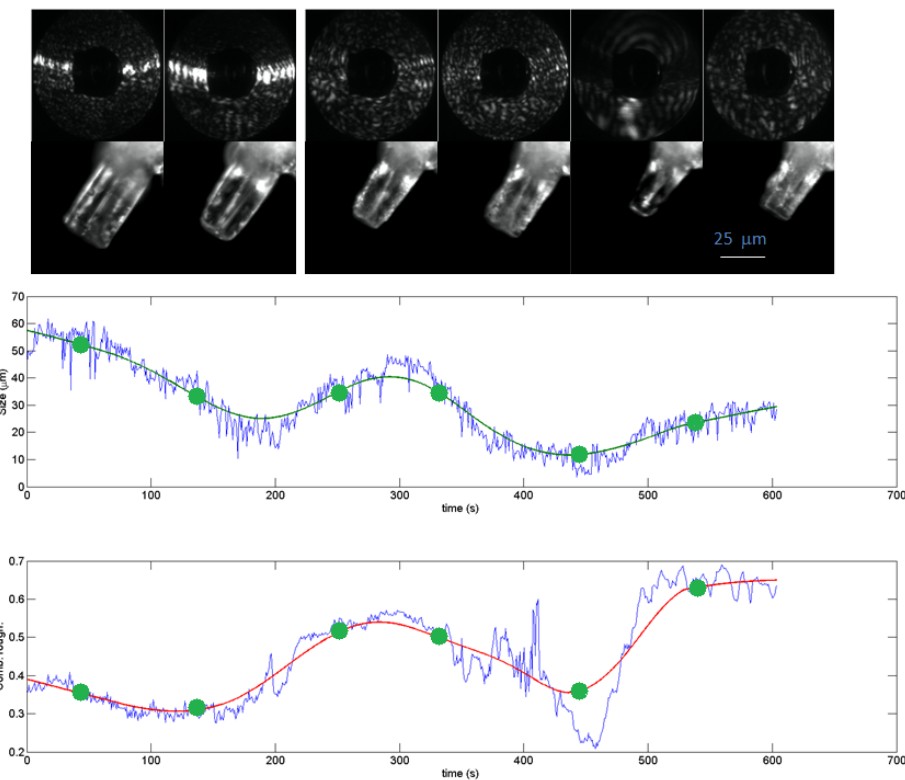

**Figure 9.** Cyclic growth-sublimation experiments with a single ice prism nucleated on a glass fibre at -30°C. After the initial growth, the saturation ratio with respect to ice was between about 1.0 and 1.05 (based on flow rate ratio, which was set to values between 0.8 l/min and 1.0 l/min wet flow, and 4.2 l/min and 4.0 l/min dry flow, see Fig. 5) and was decreased in order to partly sublimate the crystal. First row shows the 2-D scattering patterns with the corresponding time marked in green. Second row shows the ice crystal at the indicated times. The blue curves below show the actual retrieved data for particle size and roughness. The corresponding green and red best-fit curves were obtained using LOESS regression. A time-lapse video showing the entire sequence of LISA 2-D patterns is given in the supplemental material (S1), labelled with experiment time.

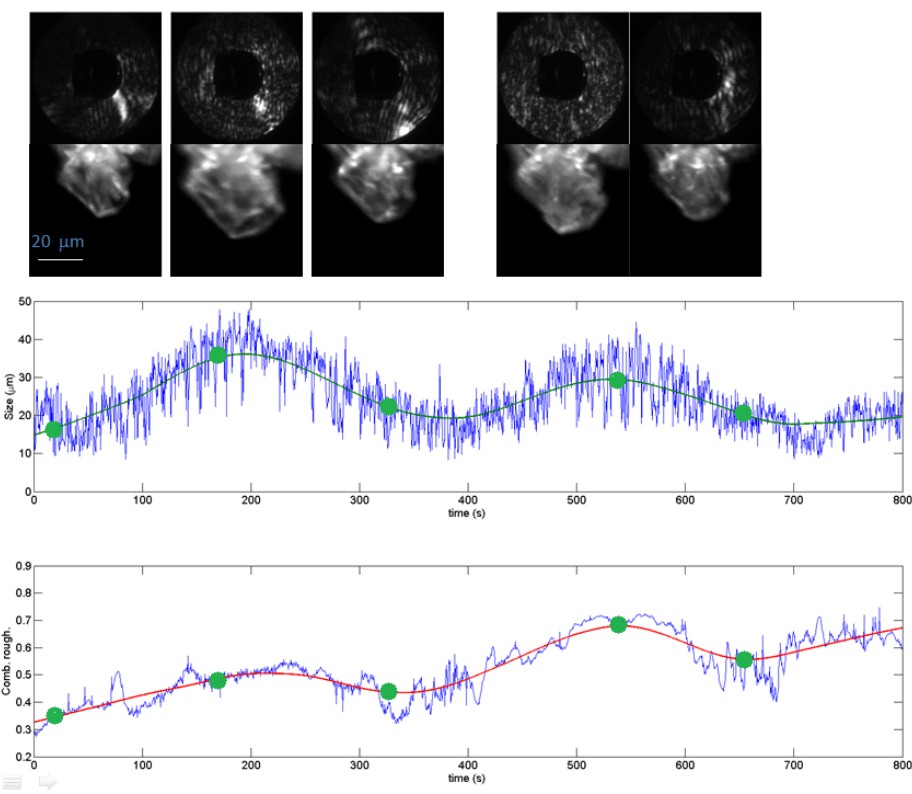

**Figure 10.** As Fig. 9 but for an ice crystal nucleated on an illite particle at -40°C. A video showing the 2-D patterns is given in the supplemental material (S2).