# Peer review of "Surface roughness during depositional growth and sublimation of ice crystals"

_Atmospheric Chemistry and Physics, 2018_

## Referee Comment (RC1) · Anonymous Referee #1 · 17 May 2018

The authors describe a new laboratory setup for investigating roughness of single ice crystals grown in the apparatus. Their main result is the appearance of a ratcheting up of ice surface roughness/irregularity as crystals are subjected to cycles of growth and sublimation, a "memory effect". Motivation for the work is given in terms of the radiative properties of ice-containing clouds in Earth's atmosphere.

I found this to be a useful and interesting contribution to what seems to be a still poorly constrained topic (ice surface roughness). The experimental apparatus and analysis methods seem to be described in sufficient detail, with a few minor exceptions (see below). The time-lapse videos provided in on-line supplement were especially valuable in aiding the interpretation of figures 10 and 11, but the manuscript stands on its own even without them.

[Figure]

OK, now for some notes, recommendations, and complaints:

1. It seems that the discussion of whether ablation leads to more or less roughening should be improved in a couple of ways. Figures 10 and 11 do seem to suggest that ablation conditions tend to reduce roughness, but I think the videos of the 2-D scattering patterns tend to tell this story more clearly. And there are more interesting patterns evident in those videos: what are the bands caused by? Finally, while the literature references given by the authors seem to point in the opposite direction, it seems worth mentioning that at at least one SEM study (Butterfield et al, Quantitative three-dimensional ice roughness from scanning electron microscopy, 2016) appears to support the authors' findings that ablating crystals tend to be less rough.

2. I found the discussion of roughening mechanisms more speculative than the authors let on. In particular, the last paragraph of section 3.1: none of what is presented in that paragraph is substantiated by evidence given in the paper. I also have a problem with the attribution at the beginning of section 3.1, in which the statement "the growth rate is slow enough for the deposited molecule to diffuse on facets to well-separated attachment sites at steps, kinks, and ledges" is not really justified, even if one can find such statements in the literature. I would point the authors to numerous studies that show that the picosecond-scale sticking coefficient on the quasi-liquid layer of ice is close to 1, and conversion of quasi-liquid to ice occurs faster than horizontal diffusion permits Neshyba et al, A quasi-liquid mediated continuum model of faceted ice dynamics, 2016, offer an alternative view. In general, I found it puzzling that there was no mention of the role of the quasi-liquid layer in these sections; if they are going to speculate, at least that factor ought to be included. At the very least, the authors should flag these sections as more highly speculative than currently indicated.

3. Mentions of "diffusion-limited" and attachment kinetics are unlikely to be understood by many ACP readers; I'd suggest elaborating a little, or omitting these points of discussion.

4. I think it would be appropriate to point the reader to the authors own prior discussion of the possibility of roughness ratcheting-up, as in Ulanowski et al, 2014.

5. Some very minor points: I think the paragraph just preceding section 3.1 is misplaced; it seems to refer to figure 8, but that figure has not been introduced yet. In section 3.2, where reference is made to "small scale vertical motions" as a possible mechanism for formation of irregular crystals, it might help to clarify that those are (I presume) atmospheric vertical motions. And there are a few misspellings here and there ("closeer" in section 3.1) that I presume will be weeded out in the next round of editing.

---

## Referee Comment (RC2) · Anonymous Referee #2 · 27 May 2018

The manuscript "Surface roughness during depositional growth and sublimation of ice crystals" by Cedric Chou et al. describes a study of ice crystal growth and sublimation performed in a new experimental setup. The experimental apparatus uses a unique combination of devices (flow diffusion tube and 2D scattering instrument) thus assuring a novelty of the results. The study is also well planned and the setup is thoroughly characterized, both by CFD modeling and experimentally. The authors demonstrate a high level of understanding of the physics behind the experiment, even if the thermodynamic parameters of the experimental system are not fully controlled. The paper is definitely worth being published, but must be thoroughly revised in many respects. I wish the paper were written more clearly. Some sections, as addressed below, require thorough editing. The relationship between the crystal evolution and its morpholog-

ical complexity is, however, convincingly demonstrated. This work should trigger off studies of this phenomena with a better control of the supersaturation and optical control of the crystal morphology. I would, however, avoid naming the effect discussed in this manuscript the "surface roughness", because this implies a quite narrow range of texture features. The sublimation and regrowth of ice crystals often create a polycrystalline aggregate of tiny crystals that can have smooth surfaces. The ultimate example is the "Bucky ball" crystals as in Baran (2012). Should such aggregate be named "rough" or "irregular" or somehow else? Would scattering patterns on such crystals be identical? I would really like to see a thorough discussion of these issues in the introduction and a clear separation of "surface roughness" from the "morphological irregularity" throughout the manuscript. In some sense, this is already done by introducing the "combined roughness" based on the 2D scattering patterns analysis. The same should be done with respect to surface texture and geometry of the crystals, and the approach suggested in this manuscript (combination of microscope observation with scattering measurements) seems to be very promising for achieving this goal. Below please find my comments which I hope would be helpful in improving the readability of the manuscript. The parts in the manuscript I am addressing are identified by page and line number and the citation are given in italic.

1. Introduction: How is the surface roughness defined and what is the quantitative measure of surface roughness? In the introductory part, the irregularity of ice crystals seems to be treated in parallel with the concept of surface roughness. However, the manuscript is titled clearly "surface roughness... ". The introduction (and the manuscript) would very much benefit from a clear definition of surface roughness as compared to habit irregularity. It would be also very helpful if you could think of a way to introduce a quantitative parameter to characterize physical surface roughness.

2. Page 2 Line 31: "In the experiments, the ice crystals are fixed within the measuring volume and exposed to thermodynamic conditions... " were there many crystals in the sample volume?

3. Page 3, line 24. "...which ensures that the 22° halo scattering from ice prisms is included." I am afraid a typical reader would not know what you are talking about. You can't expect anyone being familiar with refraction theory in hexagonal ice columns. Is this detail really needed here? The same line, replace "lower angles" with "smaller angles".

4. Page 3, line 25. "The camera images are digitized as 12-bit TIFF files..." you can't possibly mean that the camera produces analog images that have to be digitized afterward?

5. Page 3, line 30. Figure 2 does not show the fiber-optics illumination, could you show how it was located with respect to the sample volume?

6. Page 4. Section 2.1.3 "Operating principle". The content of this section does not correspond to its title. Is that the operation principle of the whole setup or the flow diffusion channel? Before describing the simulation results, please explain exactly what has been simulated and what was the purpose of the simulation (I presume, the fast control of water-ice supersaturation in the vicinity of ice crystal located in LISA).

7. Some sentences don't make sense to me: "For a sufficiently high gas flow representing the residence time of the gas flow, the thermodynamic equilibrium between the wall and the gas flow will not be reached." Does the flow represent the residence time or vice versa? Please rewrite in a clear language.

8. In Figure 3, please make the legends more clear. You should explain what the flow rates for various lines mean (total flow followed by the flow rates of dry and humidified flows at the inlet?). The green line in panel (b) has no dry/wet flow specification, why? Since the wall and inlet temperatures are the same in all panels, consider moving them into the figure caption.

9. Panel (b) of Figure 3 uses Kelvin as temperature units, but all other figures are in °C. For clarity, consider using the same units everywhere. The line showing the length

of the tube (1 m) should be present in all panels, alternatively, you could consider truncating the simulation lines at 1 m axial position.

10. Line 23: "...this can be also done on a short time scale (about 5 s) by controlling the ratio of the dry and the wet sheath air flow while the total flow is kept constant." In the beginning, you mentioned that there was no separate aerosol flow along the center of the tube, so what is the "sheath flow" for? Was the humidity of the sheath flow controlled separately? Where is this 5-second estimation coming from? Was it measured or simulated?

11. Figure 4 and discussion thereof on page 5 casts many questions in conjunction with the data of figure 6: What are the solid lines: interpolation of numerical model results or something else? Was there wet flow in the model calculations and what were the wall boundary conditions? It appears to me that the measurements have been conducted under dry conditions, without ice coating the walls of the flow tube. Is that the case? Any idea why the measurements and model calculations deviate from each other at low wall temperature? Please address these issues thoroughly.

12. If the only purpose of Figure 5 is to demonstrate that IRIS "...can be used over a broad temperature range", please consider moving it into a supplementary material. It does not contribute to instrument characterization above what has been shown in Figure 4. Besides, it is unclear what are the solid lines on the color mapping.

13. Figure 6 and the discussion thereof on page 5: for the sake of comparison, please keep the same colors as in Figure 4 (red for T_wall = -40°C and so on).

14. The deviation of measured RHi values from FLUENT results is striking, although FLUENT apparently makes a good job reproducing the flow temperature at the outlet (I am referring to the figure 4, the case of T_wall = -30°C). There, FLUENT underestimates the temperature only by 1K, which translates into 10% difference of the water vapor pressure at this temperature but not into 20% as suggested by Figure 6! Also, why don't you show FLUENT results for other wall temperatures?

15. Does the non-linearity of the measured RHi data reflect the time evolution of the flow temperature field, as discussed on page 5, starting from the line 14? Have the measurements been taken by stepwise increasing the wet flow? Would you expect a different behavior if the wet flow was decreasing instead?

16. Page 6 line 6: I believe the correct name is Gray Level Co-occurrence Matrix (GLCM). I don't know what a "co-matrix" is. What is the definition of the image "texture"? How is it different from "brightness distribution"? A typical atmospheric scientist would know little to nothing about it.

17. Page 6 line 10 and on: since the concept of GLCM and its features is the central one in the manuscript, it would be nice to include a definition of GLCM "energy" which is used in the equation 1 to calculate the "combined roughness" but is not defined anywhere in the manuscript. This is even more so because the cited paper (Ulanowski et al., ACP 2014) does not provide any explanation of "energy" either, referring to the original paper by Haralik et al, 1973. However, Haralik et al. have not used the term "energy" among the statistical descriptors of image texture. It is therefore impossible for the reader to track down the definition of the term "energy" based on the provided information. Clarification of this issue is strongly advisable.

18. A follow-up question to equation 1: The term "combined roughness" and the way it is discussed later suggest that this quantity describes both the irregularity (that is, the degree of deviation from a pristine habit) and the true physical surface roughness. Is that correct and if yes, what is their relative contributions?

19. Page 6, line 21 and on: The method of size determination should be described in much more detail as it is given in the present form. For one, it is not clear at all if the size of the ice crystal has been always determined based on the analysis of the speckle area alone. The optical setup includes a microscope and the example microscope pictures definitely show that they were good enough to determine the size of the crystals within a few micron accuracy. This, however, is not mentioned explicitly. Even if the speckle

area provides the necessary accuracy of size determination, one would certainly want to validate this method against the old-fashioned visual examination? This brings me to a question how exactly the size of the crystal has been retrieved from the speckle area? The only explanation in the manuscript is at the end of section 2.3, stating: "In addition, the size of the ice particles, which is inversely proportional to the average area of speckle spots, is retrieved". However, the citing paper (Ulanowski et al., 2012) shows clearly that the relationship between the speckle area and size does not follow the simple inverse law (their equation 12 and figure 5). If the functional form of the relationship is not known, the only possibility that is left is to construct a calibration curve from the measurements where the crystal size is retrieved independently (using, for example, the optical microscope). Could you show such a curve? What are the uncertainties of size determination based on the speckle area analysis?

20. On the other hand, the visual inspection is claimed to be used to "[. . .] compensate for temporal changes of the thermodynamic conditions caused by the ice formation at the tube wall" by adjusting the flow rate if the crystal growth slows down. These should be explained more clearly: were the microscope images used to control the growth rate of the ice crystal AND the speckle area analysis used to measure the crystal size in parallel? How do these two methods compare?

21. Page 6 line 25: "[. . .] and the amount of speckle represents crystal roughness." This is one example where the roughness should be clearly defined. Are you talking about the roughness of the surface or "combined roughness", which if I understand correctly, is the crystal irregularity plus surface roughness?

22. Section 3 Results and discussion.

23. Figure 7 is a beautiful example of the 2D interference pattern produced by smooth and rough crystals. Could you show the corresponding microscope images of the crystals responsible for them?

24. Page 7 line 26: "Fast growth can moreover lead to the creation of defects and

ionization, ..." what exactly do you mean by "ionization"? Charging, creation of the local or surface charge?

25. Section 3.1. It is stated several times in the manuscript that the supersaturation could not be determined precisely due to the instability of the thermodynamic conditions in the flow tube. You are, nevertheless, able to estimate the supersaturation with an accuracy of around $\pm 5\%$ (as in lines 3 - 4 on page 8), which is not that bad for a highly dynamic system. Given the amount of effort that has been put into characterization of the flow tube and the fact that the supersaturation is indeed the key factor controlling the morphology of the ice crystals, I would suggest that you rewrite the characterization section clearly stating the range of supersaturation and the accuracy you could achieve but avoiding saying that the supersaturation could not be controlled. This creates unnecessary distrust in your results and shifts the focus of the discussion away from the physical mechanisms of surface roughening.

26. Page 8 lines 8-9: "The crystals can be compared directly as they grow from 20 $\mu$m to 29 $\mu$m, after fitting trend curves using LOESS". What trend curves? What is LOESS? Was the size of the crystals determined from the speckle area analysis? What was the accuracy of such determination? Could you provide the confidence intervals for the "LOESS" fit? Would there be any growth in the confidence intervals for the "slow" growth case? What does "raw" in the legend of figure 8 means: measurement points, raw data? Please be more specific and more careful in presenting the results!

27. Page 8 lines 17 - 28. I support the idea that nucleation of stacking disordered ice can be responsible for the formation of irregular crystals, but how does this relate to the surface roughness? I might remind the authors again that the title of the manuscript is "Surface roughness during depositional growth..."

28. Figure 9: Please use conventional way for naming the axis. The variables "droughness" and "dsize" are not defined anywhere in the text. Besides, what size is that: radius, diameter, characteristic size...?

29. Page 10 lines 8 - 9. "Careful examination of the retrieved crystal size shown in Fig 10 indicates markedly slower growth in later cycles, despite similar supersaturation levels". To my opinion, this is stretching the imagination too far. There are only two growth cycles delivering comparable data, and the difference in the growth slope can be caused by anything else. How similar are the supersaturation values? Was the size change confirmed by optical microscope? Why does the same behavior not show up in Figure 11?

30. One more comment on this point. To my understanding, the growth rate based on the "optical size", as derived from the speckle area analysis, is directly related to the rate of growth of a volume equivalent diameter (or any other characteristic size describing the envelope dimension of the crystal). The growth rate based on such equivalent diameter is directly proportional to the mass growth rate. As "combined roughness" increases (as you have shown nicely), the ratio of surface to mass increases too, meaning that creating more surface in case of a growing complex crystal does not contribute to mass growth in the same way as in case of a growing pristine hexagonal column or plate. What implication this effect would have for the atmospheric phenomena is a question which, I am afraid, cannot be answered without thorough modeling of crystal growth with the cloud microphysical feedbacks.

31. Page 11 line 2-3: "It is very likely, as shown in our experiments, that at higher supersaturation rougher crystals will develop at the expense of smoother ones." I strongly doubt it. What would be the mechanism of such competition? Would you expect the pressure difference above smooth and rough surfaces? If not, why would rough crystals grow preferentially if both rough and smooth crystals are exposed to a supersaturated water vapor? Please clarify this statement or remove from the discussion.

32. Page 11 line 22 and on: "Finally, we note that rough ice surfaces are associated with stronger electrical charging (Caranti and Illingworth, 1983; Dash et al., 2001; Dash and Wettlaufer, 2003), hence the presence of roughness may influence storm electrification". This is indeed very interesting link that is worth discussing in more detail.

Could you say a few words explaining what mechanism underlay this phenomenon? I think this is the most far leading mechanism among other atmospheric applications.

Interactive
comment

---

## Referee Comment (RC3) · Anonymous Referee #3 · 8 Jun 2018

General Comments:

This unique laboratory study combines a laminar flow tube with a laboratory version SID-3 instrument, where flows from a "dry" and a "wet" laminar flow tube are mixed to control the supersaturation characterizing ice crystal growth at the flow tube outlet where SID-3 measurements are made (including microscope imagery). The methodology is adequately explained while the results are well explained, and the paper is well organized. The results advance our knowledge of the dependence of ice particle optical properties on ice growth/sublimation processes. I did not find much to criticize in this study.

Specific Comments:

[Figure]

1. Page 5, line 2 regarding Fig. 4: The measurements agree well with the Fluent calculations except at -40 °C at low flow rates. Please suggest reasons for these differences.

2. Page 9, lines 15-16: "these observations indicate that the more growth-sublimation cycles are performed, the rougher the crystal can become." Figure 11 does not seem to support this. Rather, the 3rd maximum in surface roughness in Fig. 11 (corresponding to the 3rd growth cycle) is slightly lower on average than the 2nd maximum in Fig. 11 (although both maximums are comparable). Therefore, it appears possible that a limiting roughness threshold exists that would not be exceeded in subsequent growth-sublimation cycles. This possibility should be acknowledged. Such a possibility seems consistent with our theoretical understanding of ice crystal surface kinetics and growth processes. Moreover, future work should explore this possibility by analyzing 3 or more continuous growth-sublimation cycles in multiple experiments at various wall temperatures. If a laboratory roughness threshold were established (possibly being supersaturation- and temperature-dependent), then the next logical step would be to look for evidence of this in natural cirrus clouds. Quantifying and bounding the degree of ice crystal surface roughness is needed to reduce uncertainty in the cirrus cloud radiative effect (CRE) in climate models.

Technical Comments:

1. Page 3, line 23: space between "the" and "central".

2. Page 16, lines 19-21: Reference cited incorrectly. Title should be "Cloud chamber experiments on the origin of ice crystal complexity in cirrus clouds", and the year of publication should be 2016. I have not checked other references; the authors should check these too.

3. Figure 3: In lower panels, the y-axis labels should be changed from "ration" to "ratio". Regarding saturation profile panel "b", should %5 l/min be 5 l/min?

[Figure]

4. Figure 4: Flow units are in "dl/min"; should this be l/min? If not, define dl.

---

## Author Response (AR1)

*Responses to Referee #1*
"Surface roughness during depositional growth and sublimation of ice crystals" *by Cèdric Chou et al.*

The authors describe a new laboratory setup for investigating roughness of single ice crystals grown in the apparatus. Their main result is the appearance of a ratcheting up of ice surface roughness/irregularity as crystals are subjected to cycles of growth and sublimation, a memory effect. Motivation for the work is given in terms of the radiative properties of ice-containing clouds in Earths atmosphere.
I found this to be a useful and interesting contribution to what seems to be a still poorly constrained topic (ice surface roughness). The experimental apparatus and analysis methods seem to be described in sufficient detail, with a few minor exceptions (see below). The time-lapse videos provided in on-line supplement were especially valuable in aiding the interpretation of figures 10 and 11, but the manuscript stands on its own even without them.

We thank the Referee for the encouraging and insightful comments, questions and helpful suggestions. We list them below, together with our clarifications and changes to the manuscript in blue.

1. It seems that the discussion of whether ablation leads to more or less roughening should be improved in a couple of ways. Figures 10 and 11 do seem to suggest that ablation conditions tend to reduce roughness, but I think the videos of the 2-D scattering patterns tend to tell this story more clearly. And there are more interesting patterns evident in those videos: what are the bands caused by? Finally, while the literature references given by the authors seem to point in the opposite direction, it seems worth mentioning that at at least one SEM study (Butterfield et al, Quantitative three-dimensional ice roughness from scanning electron microscopy, 2016) appears to support the authors findings that ablating crystals tend to be less rough.

We believe that most readers would find it hard to interpret the videos - that's why we rely on the quantitative measure of roughness instead. But to aid the interpretation, in addition to the existing explanation in section 3, we now provide further explanatory text at the end of section 2.3:

> "The presence of isolated bright spots or bands is an indication of flat crystal facets, while spots covering a large proportion of the pattern signify the presence of roughness or high complexity".

We have not cited the work of Butterfield et al. (2016) because in common with many other studies (several of which we do cite) it investigated ice growth in near vacuum, which as we argue is less relevant to atmospheric processes, and because it did not control for supersaturation, which we find to be a critical factor.

2. I found the discussion of roughening mechanisms more speculative than the authors let on. In particular, the last paragraph of section 3.1: none of what is presented in that paragraph is substantiated by evidence given in the paper. I also have a problem with the attribution at the beginning of section 3.1, in which the statement the growth rate is slow enough for the deposited molecule to diffuse on facets to well-separated attachment sites at steps, kinks, and ledges is not really justified, even if one can find such statements in the literature. I would point the authors to numerous studies that show that the picosecond-scale sticking coefficient on the quasi-liquid layer of ice is close to 1, and conversion of quasi-liquid to ice occurs faster than horizontal diffusion permits Neshyba et al, A quasi-liquid mediated continuum model of faceted ice dynamics, 2016, offer an alternative view. In general, I found it puzzling that there was no mention of the role of the quasi-liquid layer in these sections; if they are going to speculate, at least that factor ought to be included. At the very least, the authors should flag these sections as more highly speculative than currently indicated.

We admit that our discussion is speculative, not least because this initial study focuses mainly on quantifying the phenomenology, rather than the causes, so we have insufficient information at this stage to pinpoint definite origins of the observed roughness. Nevertheless, the Reviewer's comment highlights the major issue we have already pointed out: different growth studies show differences in crystal behaviour that remain to be explained, and there are many, sometimes conflicting views reported in the literature. An attempt at clarification, also pointing out the difference between experiments in air and in near vacuum, was for example provided by an interactive comment in ACP by Kiselev (2014). To reinforce this last point we now insert additional text and references in section 3.2:

> "This distinction is known to lead to different growth rates as well as habits (Beckmann, 1982; Kuroda and Gonda, 1984;".

"Furthermore, in experiments carried out under atmospherically-relevant air pressures, crystals having undergone more than one growth cycle tended to develop more faults (Beckmann, 1982)." and

"[A further difference between the diffusion limited and kinetics-limited growth is that the former can lead to] increased numbers of faults (Beckmann, 1982) and"

"Beckmann, W.: Interface kinetics of the growth and evaporation of ice single crystals from the vapour phase: III. Measurements under partial pressures of nitrogen, J. Crystal Growth, 58, 443451, doi:10.1016/0022-0248(82)90291-3, 1982.
Kuroda, T., and Gonda, T.: Rate determining processes of growth of ice crystals from the vapour phase, Part II: investigation of surface kinetic processes, J. Meteor. Soc. Jap., 62, 563572, doi:10.2151/jmsj1965.62.3_563, 1984."

For the atmospherically relevant case of growth and sublimation in air, we point out that the process is vapour diffusion limited, not limited by attachment kinetics. In this context, the timescale of the processes occurring on the surface is less critical, as more time is available for lateral diffusion. Moreover, the lateral diffusion lengths are reported to be high in this context (e.g. Pfalzgraf et al., 2011), justifying our qualitative description.

As for the relevance or otherwise of the quasi-liquid layer (QLL) concept itself, we have decided that this side topic was again too broad and complex to be discussed in our paper. Nevertheless, we can say here that while QLL appear to be important at warmer temperatures, close to the melting point, they become thinner and correspondingly less important at lower temperatures. At the temperatures relevant here the thickness of the QLL is reported to be very low, below the lattice constants of ice (e.g. Conde et al., 2008). We can also point out that much of this area is subject to a similar misapprehension as ice growth in vacuum: most QLL studies are done by molecular modelling in the absence of air, making them less relevant to atmospheric processes. This is compounded by differing use of terminology: molecular-scale roughness, much discussed in the literature, is different from the roughness on the "optical" scale (i.e. on the scale of the wavelength of light or larger) that is the subject of this work. Moreover, to our knowledge, molecular dynamics modelling studies of the QLL have so far failed to replicate the surface roughness observed experimentally; it was even argued that roughness may arise due to processes at larger scale, potentially fitting the

stacking disorder connection postulated here (Pfalzgraf et al., 2010). On a broader note, we think that it would be brave to try to overturn decades of evidence and crystal growth theory by claiming overriding importance for QLLs in this context. They do not yet explain many features of ice surface growth (e.g. growth inhibition and supersaturation thresholds). Conversely, some relevant features can be explained, somewhat counter-intuitively, by layered growth without the recourse to QLLs, e.g. rounding during sublimation (Nelson, 1998). Layered growth and sublimation, and the presence of well-defined elementary steps and terraces has been demonstrated in many systems, including ice; moreover, it can lead to larger step heights (relevant here) through step bunching (e.g. Nelson, 1998; Peterson et al., 2010; Sazaki et al., 2010; Misbah et al., 2010). So we are unsure what the relevance of QLLs is and how to bring this concept in without making the discussion more speculative (which the Reviewer advises against).

Lastly, the Neshyba et al. (2016) study authors admit that the modelling conditions do not make it directly applicable to atmospheric processes (p. 14049 therein) and do not even refer to roughness; we therefore fail to see its relevance.

To clarify the importance of step growth, we insert the following text and additional references in section 3.2.

> "Moreover, the bunching of elementary molecular steps, possibly due to the Schwoebel effect (Misbah et al., 2010), can result in the creation of larger, microscopic (as opposed to elementary) steps that can be seen in SEM micrographs (Cross, 1969).
>
> Misbah, C., Pierre-Louis, O., and Saito, Y.: Crystal surfaces in and out of equilibrium: A modern view, Rev. Mod. Phys., 82, 981-1040, doi:10.1103/RevModPhys.82.981, 2010."

3. Mentions of diffusion-limited and attachment kinetics are unlikely to be understood by many ACP readers; I'd suggest elaborating a little, or omitting these points of discussion.

At the first mention we now additionally clarify that we refer to vapour diffusion. The distinction we point out is important, as it permits an evaluation of the (ir)relevance of various laboratory experiments to the atmospheric context, and differences between their outcomes. We have already provided several relevant references to kinetics- and diffusion-limited growth at several points in the article for interested readers to follow.

> "...growth in the absence of air that takes place in a SEM chamber, instead of being limited by vapour diffusion as is the case for ice at tropospheric pressures, becomes limited by the attachment kinetics..."

4. I think it would be appropriate to point the reader to the authors own prior discussion of the possibility of roughness ratcheting-up, as in Ulanowski et al, 2014.

At the beginning of section 3.2 we now refer the readers to the prior discussion in the reference suggested.

> "Ulanowski et al., 2014"

5. Some very minor points: I think the paragraph just preceding section 3.1 is misplaced; it seems to refer to figure 8, but that figure has not been introduced yet. In section 3.2, where reference is made to small scale vertical motions as a possible mechanism for formation of irregular crystals, it might help to clarify that those are (I presume) atmospheric vertical motions. And there are a few misspellings here and there (closeer" in section 3.1) that I presume will be weeded out in the next round of editing.

We have placed this explanatory text where it is because it refers to several subsequent figures. Concerning vertical motions, we insert the word "atmospheric" for clarification.

References for Reply:

Conde, M. M., Vega, C., and Patrykiejew, A.: The thickness of a liquid layer on the free surface of ice as obtained from computer simulation, J. Chem. Phys., 129, 014702, doi:10.1063/1.2940195, 2008.

Kiselev, A.: Interactive comment on "Mesoscopic surface roughness of ice crystals pervasive across a wide range of ice crystal conditions" by N. B. Magee et al., Atmos. Chem. Phys. Discuss., 14, C4758-C4763, http://www.atmos-chem-phys-discuss.net/14/C4758/2014, 2014.

Sazaki, G., Zepeda, S., Nakatsubo, S., Yokoyama, E., and Furukawa, Y.: Elementary steps at the surface of ice crystals visualized by advanced optical microscopy, Proc. Nat. Acad. Sci., 107, 19702-19707, doi:10.1073/pnas.1008866107, 2010.

Peterson, H., Bailey, M., and Hallett, J.: Ice particle growth under conditions of the upper troposphere, Atm. Res., 97, 446-449, doi:10.1016/j.atmos res.2010.05.013, 2010.

*Responses to Referee #2*
"Surface roughness during depositional growth and sublimation of ice crystals" *by Cèdric Chou et al.*

The manuscript Surface roughness during depositional growth and sublimation of ice crystals by Cedric Chou et al. describes a study of ice crystal growth and sublimation performed in a new experimental setup. The experimental apparatus uses a unique combination of devices (flow diffusion tube and 2D scattering instrument) thus assuring a novelty of the results. The study is also well planned and the setup is thoroughly characterized, both by CFD modeling and experimentally. The authors demonstrate a high level of understanding of the physics behind the experiment, even if the thermodynamic parameters of the experimental system are not fully controlled. The paper is definitely worth being published, but must be thoroughly revised in many respects. I wish the paper were written more clearly. Some sections, as addressed below, require thorough editing. The relationship between the crystal evolution and its morpholog-ical complexity is, however, convincingly demonstrated. This work should trigger off studies of this phenomena with a better control of the supersaturation and optical control of the crystal morphology. I would, however, avoid naming the effect discussed in this manuscript the surface roughness, because this implies a quite narrow range of texture features. The sublimation and regrowth of ice crystals often create a polycrystalline aggregate of tiny crystals that can have smooth surfaces. The ultimate example is the Bucky ball crystals as in Baran (2012). Should such aggregate be named rough or irregular or somehow else? Would scattering patterns on such crystals be identical? I would really like to see a thorough discussion of these issues in the introduction and a clear separation of surface roughness from the morphological irregularity throughout the manuscript. In some sense, this is already done by introducing the combined roughness based on the 2D scattering patterns analysis. The same should be done with respect to surface texture and geometry of the crystals, and the approach suggested in this manuscript (combination of microscope observation with scattering measurements) seems to be very promising for achieving this goal. Below please find my comments which I hope would be helpful in improving the readability of the manuscript. The parts in the

manuscript I am addressing are identified by page and line number and the citation are given in italic.

We thank the Referee for a very thorough and encouraging review and many insightful comments, questions and helpful suggestions that allowed us to improve the manuscript. We list them below, together with our clarifications (in blue) and changes to the manuscript.

1. Introduction: How is the surface roughness defined and what is the quantitative measure of surface roughness? In the introductory part, the irregularity of ice crystals seems to be treated in parallel with the concept of surface roughness. However, the manuscript is titled clearly surface roughness... . The introduction (and the manuscript) would very much benefit from a clear definition of surface roughness as compared to habit irregularity. It would be also very helpful if you could think of a way to introduce a quantitative parameter to characterize physical surface roughness.

We state clearly that fine surface roughness and large-scale irregularity are treated together. Further discussion of the meaning and significance of the measure of roughness goes beyond the scope of present work and is dealt with in detail in the cited articles (Ulanowski et al., 2012, 2014). Among others, it was pointed out that the lack of distinction between "roughness" and "complexity" from the point of view of 2-D scattering is likely to also apply to other light scattering properties. So the distinction may be to some degree artificial. Nevertheless, crystals in the experiments were relatively simple prisms, not complex ones, so the study is more directly relevant to surface roughness, hence our choice of emphasis.

2. Page 2 Line 31: In the experiments, the ice crystals are fixed within the measuring volume and exposed to thermodynamic conditions... were there many crystals in the sample volume?

We insert the text ", generally single,":

> "In the experiments, the ice crystals, generally single, are fixed within the measuring volume and exposed to thermodynamic conditions simulating single or multiple growth cycles at various temperature and saturation ratio."

3. Page 3, line 24. ...which ensures that the 22? halo scattering from ice prisms is included. I am afraid a typical reader would not know what you are talking about. You cant expect anyone being familiar with refraction theory

in hexagonal ice columns. Is this detail really needed here? The same line, replace lower angles with smaller angles.

We alter the text to say "bright feature associated with the familiar halo occurring for ice prisms at the scattering angle of 22°". We do replace "lower angles with "smaller angles", as suggested.

4. Page 3, line 25. The camera images are digitized as 12-bit TIFF files... you can't possibly mean that the camera produces analog images that have to be digitized afterward?

That is indeed what happens internally to the camera.

5. Page 3, line 30. Figure 2 does not show the fiber-optics illumination, could you show how it was located with respect to the sample volume?

The text mentions the illumination; we now add that it is "in-line" with the microscope tube and add extra detail in Fig. 2.

6. Page 4. Section 2.1.3 Operating principle. The content of this section does not correspond to its title. Is that the operation principle of the whole setup or the flow diffusion channel? Before describing the simulation results, please explain exactly what has been simulated and what was the purpose of the simulation (I presume, the fast control of water-ice supersaturation in the vicinity of ice crystal located in LISA).

We create instead a new section entitled "Numerical simulations and thermodynamic characterisation". Out statement concerning the purpose of the simulations was probably too short. We change the text substantially as given below. The purpose of the steady state simulations was not to show the fast control of water-ice supersaturation. For that a transient model would be required. The response time of the system was a) calculated from mass flow rates and total volume of the system, and b) measured.

> "The thermodynamic conditions at the tube outlet were extensively studied by means of computational fluid dynamics (CFD) simulations of the laminar flow tube, and by measurements of flow velocity, temperature and dew point at the tube outlet. Both the numerical simulations and the measurements have been done to characterise the experimental setup, as well as to demonstrate the fast control of temperature and supersaturation in the measuring volume.

The numerical simulations were done with the commercially available CFD code Fluent (Ansys Inc., USA). The Fluent model is a general purpose FVM (finite volume method) CFD model allowing the simulation of a wide range of small scale fluid flow problems. Here, the flow through the flow tube was simulated including a multicomponent treatment of the flow. The model accounts for the coupled processes of mass and heat transfer. With respect to the geometry and the laminar flow regime, the simulations were done on a 2-dimensional axisymmetric Cartesian grid by means of a pressure based steady state solver. Additional information about the numerical model, which has already been successfully applied to the characterisation of the laminar flow tube LACIS, can be found for example in Stratmann et al. (2004); Voigtländer et al. (2004); Voigtländer (2007) and Hartmann at al. (2011).
To illustrate the operating principle of the laminar flow diffusion channel, calculated thermodynamic profiles along the tube axis are shown in Fig. 3. ..."

7. Some sentences don't make sense to me: For a sufficiently high gas flow representing the residence time of the gas flow, the thermodynamic equilibrium between the wall and the gas flow will not be reached. Does the flow represent the residence time or vice versa? Please rewrite in a clear language.

We modify the paragraph by adding the text:

"If the residence time of the gas flow (controlled by the mass flow rate) is large enough, the gas flow cools down until thermodynamic equilibrium with the tube wall is reached. Conversely, for a sufficiently fast flow equilibrium will not be reached."

8. In Figure 3, please make the legends more clear. You should explain what the flow rates for various lines mean (total flow followed by the flow rates of dry and humidified flows at the inlet?). The green line in panel (b) has no dry/wet flow specification, why? Since the wall and inlet temperatures are the same in all panels, consider moving them into the figure caption.
9. Panel (b) of Figure 3 uses Kelvin as temperature units, but all other figures are in °C. For clarity, consider using the same units everywhere. The line showing the length of the tube (1 m) should be present in all panels, alternatively, you could consider truncating the simulation lines at 1 m axial

position.

We change the figures according to the suggestions of the reviewer.

10. Line 23: ...this can be also done on a short time scale (about 5 s) by controlling the ratio of the dry and the wet sheath air flow while the total flow is kept constant. In the beginning, you mentioned that there was no separate aerosol flow along the center of the tube, so what is the sheath flow for? Was the humidity of the sheath flow controlled separately? Where is this 5-second estimation coming from? Was it measured or simulated?

We agree with the reviewer that the statement of "sheath air" could be misleading. The reviewer is right, there is no separate aerosol flow along the tube center. There is only one particle-free gas flow along the tube. We delete the word "sheath".
The 5 s estimation comes from a calculation, but was also measured by observing the ice crystals. For the calculation, we simply considered the total volume of the system downstream of the humidifier (not only the flow tube) and the flow rates.

11. Figure 4 and discussion thereof on page 5 casts many questions in conjunction with the data of figure 6: What are the solid lines: interpolation of numerical model results or something else? Was there wet flow in the model calculations and what were the wall boundary conditions? It appears to me that the measurements have been conducted under dry conditions, without ice coating the walls of the flow tube. Is that the case? Any idea why the measurements and model calculations deviate from each other at low wall temperature? Please address these issues thoroughly.

The solid lines in Fig. 4 and Fig. 6 are an interpolation of the experimental data - we add a note in the caption of the Fig. It is correct that there was no wet flow for the temperature characterisation data shown in Fig. 4. This means, the investigations have been done under dry conditions (by using pressurized air with a dew point slightly below -40°C). Using dry conditions holds for both, numerical simulations and measurements. The wall boundary condition in the simulations was zero flux. The reason for determining the temperature at dry conditions was that temperature measurements are not trivial under cold and wet conditions. For example, icing at the temperature sensor might occur. We know from wet simulations (with $S = 1$ at the wall boundary) that the temperature at the tube outlet is not significantly influenced by the presence of water in the system.

Furthermore, we also think that the differences between experimental data and simulation results are caused at least partly by the measurement technique. Especially at low flow velocities the temperature sensor, which was positioned in the optical measuring volume of LISA, several millimeters below the tube outlet, might not give true values. We spent much time using different types of sensors (various Pt100 and thermocouple sensors) to find out which one gives the best results for our application. In conclusion, even if the sensor is precisely calibrated (e.g. in an ethanol bath against a reference PT100 sensor), the difference between measurements and simulation results is probably due to technical measurement issues.
Measurements and simulations shown in Fig. 6 have been done for wet flow conditions. To minimize the effect of icing at the walls, the measurements were done only for a short time after the wet flow was switched on.

12. If the only purpose of Figure 5 is to demonstrate that IRIS ...can be used over a broad temperature range, please consider moving it into a supplementary material. It does not contribute to instrument characterization above what has been shown in Figure 4. Besides, it is unclear what are the solid lines on the color mapping.

We follow the suggestion of the Referee and move Fig. 5 into the supplement. We also add a figure to the supplement showing flow speed (measurements and simulations) data at the tube outlet. We change the text accordingly, delete the corresponding sentences (p. 5, l. 2-4) and add instead :

> "Additionally, an extended data set of temperature measurements is shown in the supplement material."

and in the previous section we change the last sentence to (p. 4, l. 31):

> "Measured and calculated flow velocities were found to be very similar (see supplement material)."

13. Figure 6 and the discussion thereof on page 5: for the sake of comparison, please keep the same colors as in Figure 4 (red for T_wall = -40°C and so on).

There isn't one to one correspondence between the conditions in the two figures, hence the different colours. However, we followed the suggestion of the Reviewer and improved the Figs. for the revised version.

14. The deviation of measured RHi values from FLUENT results is striking,

although FLUENT apparently makes a good job reproducing the flow temperature at the outlet (I am referring to the figure 4, the case of T_wall = -30°C). There, FLUENT underestimates the temperature only by 1K, which translates into 10% difference of the water vapor pressure at this temperature but not into 20% as suggested by Figure 6! Also, why dont you show FLUENT results for other wall temperatures?

As stated in the reply to Point 6, the purpose of the simulations was to design the experiments. Therefore, the model was simplified in several ways. We reduced the flow simulations to a multicomponent (water + air), but single phase (gaseous phase) problems. Water phase transition was considered as a sink only. This means, the growing ice layer at the tube wall was not simulated. In the simulations, the wall boundary condition was defined by setting the walls to saturated conditions with respect to ice (100% at the wall temperature). Hence we are not able to simulate the temporal change of the saturation (and temperature) profile. However, one can imagine that a growing ice layer may act as an insulator increasing the temperature gradient. Additionally, the dry air flow was considered to be completely dry in the simulations (water mass fraction of zero), while the dew point of our pressurized air was in reality between about -50°C and -40°C. Simulation results are shown and compared to experimental data for the case of Twall = -30°C because most the the experiments have been done at this temperature. For the data shown here, it is therefore the most relevant temperature value. Furthermore, measurements are getting more and more challanging at even lower temperatures. As stated above, temperature measurements at -40°C might be biased due to technical limitations. The same holds for dew point measurements. Therefore, we would expect that the differences between experiments and simulations results are a even bigger at -40°C.

15. Does the non-linearity of the measured RHi data reflect the time evolution of the flow temperature field, as discussed on page 5, starting from the line 14? Have the measurements been taken by stepwise increasing the wet flow? Would you expect a different behavior if the wet flow was decreasing instead?

Yes, one reason for the non-linearity of the measured RHi data is the time evolution of the temperatrue field. A second reason is the wall loss of water vapour at high RH. Water vapour is transported to the tube wall forming an ice layer. This sink also depresses the supersaturation. And yes, it is correct that the measurements were taken by increasing the wet flow.
The temporal change of the temperature is not linear. We observed faster

changes at the beginning of the wall ice formation. At high RHi ice is quickly formed at the wall. Therefore, at high RHi there is not much difference if the measurements starts with a high wet flow. However, at low RHi it is. We did measurements with decreasing wet flow showing a difference.

16. Page 6 line 6: I believe the correct name is Gray Level Co-occurrence Matrix (GLCM). I dont know what a co-matrix is. What is the definition of the image texture? How is it different from brightness distribution? A typical atmospheric scientist would know little to nothing about it.

The Reviewer is right, while the term co-matrix is often used for brevity in the literature, it is more correct to say "Co-occurrence Matrix"; we make a change in the text. See also response to Point 1.

17. Page 6 line 10 and on: since the concept of GLCM and its features is the central one in the manuscript, it would be nice to include a definition of GLCM energy which is used in the equation 1 to calculate the combined roughness but is not defined anywhere in the manuscript. This is even more so because the cited paper (Ulanowski et al., ACP 2014) does not provide any explanation of energy either, referring to the original paper by Haralik et al, 1973. However, Haralik et al. have not used the term energy among the statistical descriptors of image texture. It is therefore impossible for the reader to track down the definition of the term energy based on the provided information. Clarification of this issue is strongly advisable.

Incorrect: Haralick et al. do define "energy", just that this initial paper does not call it such. In any case, readers wishing to implement the measures can easily locate literally thousands of relevant references. However, for clarification we insert the text:

"(also known as uniformity, or angular second moment)"

18. A follow-up question to equation 1: The term combined roughness and the way it is discussed later suggest that this quantity describes both the irregularity (that is, the degree of deviation from a pristine habit) and the true physical surface roughness. Is that correct and if yes, what is their relative contributions?

This question cannot be answered at this stage, as the properties are not separable and anyway cannot be defined unambiguously. Further discussion is in the reference cited.

19. Page 6, line 21 and on: The method of size determination should be described in much more detail as it is given in the present form. For one, it is not clear at all if the size of the ice crystal has been always determined based on the analysis of the speckle area alone. The optical setup includes a microscope and the example microscope pictures definitely show that they were good enough to determine the size of the crystals within a few micron accuracy. This, however, is not mentioned explicitly. Even if the speckle area provides the necessary accuracy of size determination, one would certainly want to validate this method against the old-fashioned visual examination? This brings me to a question how exactly the size of the crystal has been retrieved from the speckle area? The only explanation in the manuscript is at the end of section 2.3, stating: In addition, the size of the ice particles, which is inversely proportional to the average area of speckle spots, is retrieved. However, the citing paper (Ulanowski et al., 2012) shows clearly that the relationship between the speckle area and size does not follow the simple inverse law (their equation 12 and figure 5). If the functional form of the relationship is not known, the only possibility that is left is to construct a calibration curve from the measurements where the crystal size is retrieved independently (using, for example, the optical microscope). Could you show such a curve? What are the uncertainties of size determination based on the speckle area analysis?

The method is described in detail and validated in the reference cited, which also includes a calibration curve. However, for clarification we insert the text:

> "and is used thoughout the present work to determine crystal size. The size measured in this way represents the diameter of equal area circle projected along the line parallel to the laser beam."

See also response to Points 20 and 26.

20. On the other hand, the visual inspection is claimed to be used to [...] compensate for temporal changes of the thermodynamic conditions caused by the ice formation at the tube wall by adjusting the flow rate if the crystal growth slows down. These should be explained more clearly: were the microscope images used to control the growth rate of the ice crystal AND the speckle area analysis used to measure the crystal size in parallel? How do these two methods compare?

Since the characterization of the thermodynamic conditions was not precise enough to establish the point of equilibrium between the crystal and the vapour, we had to observe the crystal to find out whether it was growing or not, i.e. if it was in equilibrium. This established the point of reference. But otherwise the images were not used for adjusting the conditions. To clarify, we replace the last, potentially misleading sentence with the text:

> "In this way the settings corresponding to the point of equilibrium between the crystal and the vapour can be found, to act as a reference point."

The two methods cannot be directly compared because the crystal dimensions "seen" by the two methods are orthogonal (one is parallel, the other perpendicular to the laser beam).

21. Page 6 line 25: [...] and the amount of speckle represents crystal roughness. This is one example where the roughness should be clearly defined. Are you talking about the roughness of the surface or combined roughness, which if I understand correctly, is the crystal irregularity plus surface roughness?

This was addressed previously, the roughness measure combines both. And the statement cited is a qualitative explanation, simply aiding the reader in the interpretation of unfamilar 2-D scattering patterns.

22. Section 3 Results and discussion.

23. Figure 7 is a beautiful example of the 2D interference pattern produced by smooth and rough crystals. Could you show the corresponding microscope images of the crystals responsible for them?

While we agree with the Reviewer that the inclusion of images with Fig. 7 would be interesting, optical microscopy images (or cloud probe ones) do not reveal sufficient detail of surface roughness. Anyway, in this case parallel microscopy was not obtained, as the patterns were generated in a conventional cloud chamber. We now add an explanatory sentence in the caption:

> "The patterns were produced using the SID3 instrument in the AIDA cloud chamber during growth at low (left) and high (right) supersaturation (Schnaiter et al., 2016)."

24. Page 7 line 26: Fast growth can moreover lead to the creation of defects and ionization, ... what exactly do you mean by ionization? Charging, creation of the local or surface charge?

Please see the multiple references cited, which deal with this broad topic, as well as the Conclusions and the Reviewer's own Point 32.

25. Section 3.1. It is stated several times in the manuscript that the supersaturation could not be determined precisely due to the instability of the thermodynamic conditions in the flow tube. You are, nevertheless, able to estimate the supersaturation with an accuracy of around $\pm 5\%$ (as in lines 3 - 4 on page 8), which is not that bad for a highly dynamic system. Given the amount of effort that has been put into characterization of the flow tube and the fact that the supersaturation is indeed the key factor controlling the morphology of the ice crystals, I would suggest that you rewrite the characterization section clearly stating the range of supersaturation and the accuracy you could achieve but avoiding saying that the supersaturation could not be controlled. This creates unnecessary distrust in your results and shifts the focus of the discussion away from the physical mechanisms of surface roughening.

We did not say that supersaturation could not be controlled. In section 3.1 we state that "supersaturation ... could not be determined precisely". However, this may be misleadin, so we now say "directly" instead, changing the sentence to:

> "Since the supersaturation controls the growth rate but could not be determined directly with high accuracy in our experiments, ..."

26. Page 8 lines 8-9: The crystals can be compared directly as they grow from 20 $\mu$m to 29 $\mu$m, after fitting trend curves using LOESS. What trend curves? What is LOESS? Was the size of the crystals determined from the speckle area analysis? What was the accuracy of such determination? Could you provide the confidence intervals for the LOESS fit? Would there be any growth in the confidence intervals for the slow growth case? What does raw in the legend of figure 8 means: measurement points, raw data? Please be more specific and more careful in presenting the results!

LOESS is a well-established numerical technique, and is described and referenced earlier in the text. As for size, see Points 19 and 20. Concerning the "confidence intervals", these would not carry any information relevant to the behaviour of the observed crystal, as they are the outcome of sec-

ondary noise sources such as digitized image noise or mechanical vibrations, as pointed out in section 3.

27. Page 8 lines 17 - 28. I support the idea that nucleation of stacking disordered ice can be responsible for the formation of irregular crystals, but how does this relate to the surface roughness? I might remind the authors again that the title of the manuscript is Surface roughness during depositional growth...

Please see response to Point 1, it is stressed several times that we do not distinguish between fine and coarse. roughness.

28. Figure 9: Please use conventional way for naming the axis. The variables droughness and dsize are not defined anywhere in in the text. Besides, what size is that: radius, diameter, characteristic size...?

The axes are defined as rates in the caption, and the variables "roughness" and "size" in the text, see also response to Point 19 above. However, for clarity we alter the axis labels to "d(roughness)/d(time)" and "d(size)/d(time)".

29. Page 10 lines 8 - 9. Careful examination of the retrieved crystal size shown in Fig 10 indicates markedly slower growth in later cycles, despite similar supersaturation levels. To my opinion, this is stretching the imagination too far. There are only two growth cycles delivering comparable data, and the difference in the growth slope can be caused by anything else. How similar are the supersaturation values? Was the size change confirmed by optical microscope? Why does the same behavior not show up in Figure 11?

The supersaturations were the same. We bring this observation to the readers attention as it is potentially important, and we do not claim it as a "fact", hedging our bets with words like "appears to" etc. However, we cite similar behaviour observed in other systems. Moreover, concerning Fig. 11, we beg to disagree, as similar behaviour can be seen in crystal growth rate. So we are prepared to stand by our statements, and further work will confirm or contradict them. As for the microscopy, see response to Point 20.

30. One more comment on this point. To my understanding, the growth rate based on the optical size, as derived from the speckle area analysis, is directly related to the rate of growth of a volume equivalent diameter (or any other characteristic size describing the envelope dimension of the crystal). The growth rate based on such equivalent diameter is directly

proportional to the mass growth rate. As combined roughness increases (as you have shown nicely), the ratio of surface to mass increases too, meaning that creating more surface in case of a growing complex crystal does not contribute to mass growth in the same way as in case of a growing pristine hexagonal column or plate. What implication this effect would have for the atmospheric phenomena is a question which, I am afraid, cannot be answered without thorough modeling of crystal growth with the cloud microphysical feedbacks.

No, it is not "volume equivalent diameter" - see Point 19. However, we are happy to support the rest of the Reviewer's comment.

31. Page 11 line 2-3: It is very likely, as shown in our experiments, that at higher supersaturation rougher crystals will develop at the expense of smoother ones. I strongly doubt it. What would be the mechanism of such competition? Would you expect the pressure difference above smooth and rough surfaces? If not, why would rough crystals grow preferentially if both rough and smooth crystals are exposed to a supersaturated water vapor? Please clarify this statement or remove from the discussion.

We merely reiterate that our experiments show higher roughness at high supersaturation, and that this is also likely to occur in the atmosphere. However, to avoid misunderstanding, we change the words "develop at the expense of" to "tend to develop instead of".

32. Page 11 line 22 and on: Finally, we note that rough ice surfaces are associated with stronger electrical charging (Caranti and Illingworth, 1983; Dash et al., 2001; Dash and Wettlaufer, 2003), hence the presence of roughness may influence storm electrification. This is indeed very interesting link that is worth discussing in more detail. Could you say a few words explaining what mechanism underlay this phenomenon? I think this is the most far leading mechanism among other atmospheric applications.

This is indeed an intriguing possibility, that is why we speculatively mention it. However, we consider wider discussion to be beyound the scope of the present work, and we instead refer the readers to several references cited in sections 3.1 and 3.3.

*Responses to Referee #3*
"Surface roughness during depositional growth and sublimation of ice crystals" *by Cèdric Chou et al.*

This unique laboratory study combines a laminar flow tube with a laboratory version SID-3 instrument, where flows from a dry and a wet laminar flow tube are mixed to control the supersaturation characterizing ice crystal growth at the flow tube outlet where SID-3 measurements are made (including microscope imagery). The methodology is adequately explained while the results are well explained, and the paper is well organized. The results advance our knowledge of the dependence of ice particle optical properties on ice growth/sublimation processes. I did not find much to criticize in this study.

We thank the Referee for the positive comments and suggestions. Below we list our response to the two main comments.

Specific Comments:
1. Page 5, line 2 regarding Fig. 4: The measurements agree well with the Fluent calculations except at -40°C at low flow rates. Please suggest reasons for these differences.

We think that the reason for the deviation between measurements and simulations at low temperatures and (especially) low flow rates are caused by the measurement technique. Accurate temperature measurement in a gas flow under a small flow tube at low flow temperature and flow speed is not trivial. The temperature sensor, which was positioned in the optical measuring volume of LISA several millimeters below the tube outlet, might not give accurate values if the flow velocity is too small, especially at low temperatures. We spent a lot of time using different types of sensors (various Pt100 and thermocouple sensors) to find out which one gives the best results for our application. In conclusion, even if the sensor is precisely calibrated in an ethanol bath against a reference Pt100 sensor, the difference between measurements and simulation results is probably due to technical measurement issues.

2. Page 9, lines 15-16: these observations indicate that the more growth-sublimation cycles are performed, the rougher the crystal can become. Figure 11 does not seem to support this. Rather, the 3rd maximum in surface roughness in Fig. 11 (corresponding to the 3rd growth cycle) is slightly lower on average than the 2nd maximum in Fig. 11 (although both maximums are comparable). Therefore, it appears possible that a limiting roughness threshold exists that would not be exceeded in subsequent growth-sublimation cycles. This possibility should be acknowledged. Such a possibility seems consistent with our theoretical understanding of ice crystal surface kinetics and growth processes. Moreover, future work should explore this possibility by analyzing 3 or more continuous growth-sublimation cycles in multiple experiments at various wall temperatures. If a laboratory roughness threshold were established (possibly being supersaturation- and temperature-dependent), then the next logical step would be to look for evidence of this in natural cirrus clouds. Quantifying and bounding the degree of ice crystal surface roughness is needed to reduce uncertainty in the cirrus cloud radiative effect (CRE) in climate models.

We stopped the experiments after a few growth-sublimation cycles because the gross shape of the ice crystals often slightly changes with each cycle. Here, we didn't wish to mix the effects of surface roughness and larger irregularities and therefore stopped when the ice crystal started to develop significantly different morphology. So we generally agree with the Reviewer's point that there appears to be an upper limit of the (combined) roughness value, but higher values could be reached in principle during longer experiments. We consider this suggestion and will try to address this point in future investigations.

Technical Comments:
1. Page 3, line 23: space between the and central.
2. Page 16, lines 19-21: Reference cited incorrectly. Title should be Cloud chamber experiments on the origin of ice crystal complexity in cirrus clouds, and the year of publication should be 2016. I have not checked other references; the authors should check these too.
3. Figure 3: In lower panels, the y-axis labels should be changed from ration to ratio. Regarding saturation profile panel b, should %5 l/min be 5 l/min?
4. Figure 4: Flow units are in dl/min; should this be l/min? If not, define dl.

We improve the Figs. as suggested and have rechecked the references.

**Surface roughness during depositional growth and sublimation of ice crystals,* by Cedric Chou et al.**

**List of major changes**

Key: insertions are in blue, deletions are . The page and line numbers refer to the original discussion paper.

**2.1 Experimental setup**

P.2 line 31: In the experiments, the ice crystals, generally single, are fixed within the measuring volume

P.3 line 28: via an Infinitube right-angle adaptor with fibre-optics in-line illumination

**2.1.1 Laminar flow tube**

P.3 line 10: mass flow controllers ( Brooks 5850s, Brooks Instruments, Hatfield, PA, USA)

**2.1.2 Optical system**

P.3 line 23: 2-D scattering patterns are collected by LISA via an intensified CCD camera at scattering angles from 6 to 25º in an annular shape, which ensures that the bright feature associated with the familiar halo occurring for ice prisms at the scattering angle of 22º  is included. The  smaller angles in the central area are not captured due to the presence of a beam stop.
* * *
**2.2 Numerical simulations and thermodynamic characterisation**

P.4 line 2:  The thermodynamic conditions at the tube outlet were extensively studied by means of computational fluid dynamics (CFD) 5 simulations of the laminar flow tube, and by measurements of flow velocity, temperature and dew point at the tube outlet. Both the numerical simulations and the measurements have been done to characterise the experimental setup, as well as to demonstrate the fast control of temperature and supersaturation in the measuring volume.
    The numerical simulations were done with the commercially available CFD code Fluent (Ansys Inc., USA). The Fluent model is a general purpose FVM (finite volume method) CFD model allowing the simulation of a wide range of small scale 10 fluid flow problems. Here, the flow through the flow tube was simulated including the coupled processes of mass and heat transfer. With respect to the geometry and the laminar flow regime, the simulations were done on a 2-dimensional axisymmetric Cartesian grid by means of a pressure based steady state solver. With respect to the geometry and the laminar flow regime, the simulations were done on a 2-dimensional axisymmetric Cartesian grid by means of a pressure based steady state solver. Additional information about the numerical model, which has already been successfully applied to the characterisation of the laminar flow tube LACIS, can be found for example in Stratmann et al. (2004); Voigtländer et al. (2007) and Hartmann et al. (2011).
    To illustrate the operating principle of the laminar flow diffusion channel, calculated thermodynamic profiles along the tube axis are shown in Fig. 3.

 If the residence time of the gas flow (controlled by the mass flow rate) is large enough, the gas flow cools down until thermodynamic equilibrium with the tube wall is reached. Conversely, for a sufficiently fast flow equilibrium  will not be reached.

P.4 line 23: by controlling the ratio of the dry and the wet  air flow while the total flow is kept constant.

P.4 line 30: Measured and calculated flow velocities were found to be very similar ( see supplement material).

P.5 line 2:  Additionally, an extended data set of temperature measurements is shown in the supplement material.

P.5 line 26: This is done by  observation of the ice crystal with the optical microscope. Since the ice crystal growth process is highly sensitive to the prevailing thermodynamic conditions, i.e. the saturation 5 ratio determines the ice crystal growth rate, the  massflowcontrollers can be adjusted according to the microscope images. In  this way the settings corresponding to the point of equilibrium between the crystal and the vapour can be found, to act as a reference point.

**2.4 Scattering pattern analysis**

P. 6 line 7: In brief, image texture can be retrieved by using the  Grey-Level Co-occurrence Matrix (GLCM)

P. 6 line 10: among the four features of the GLCM (contrast, correlation homogeneity and energy, also known as uniformity or angular second moment),

P. 6 line 16: where  $E$ is the energy derived from the GLCM and  $K$ the kurtosis.  The combined measure is dependent on the number of independent "scattering centres"

P. 6 line 20: In addition, the size of the ice particles, which is inversely proportional to the average area of speckle spots, is retrieved and is used throughout the present work to determine crystal size. The size measured in this way represents the diameter of equal area circle projected along the line parallel to the laser beam (Ulanowski et al., 2012).

P. 6 line 22: The presence of isolated bright spots or bands is an indication of flat crystal facets, while spots covering a large proportion of the pattern signify the presence of roughness or high

**3 Results and discussion**

P. 6 line 28: In the following,  experiments are presented and discussed first, addressing two aspects the influence of supersaturation, and of regrowth cycles on the ice crystal surface roughness measures. However, we note that in general other factors may also influence crystal morphology

P. 7 line 8: Fig. 6 shows examples of  2-D scattering patterns  from smooth and rough ice columns to illustrate what patterns could be classified as originating from smooth or rough ice crystals in the following discussion.

**3.1 Slow and fast growth**

P. 7 line 26: Fast growth can moreover lead to the creation of defects and ionization, which further promote irregular growth (Beckmann, 1982; Dash et al., 2001; Dash and Wettlaufer, 2003; Pantaraks and Flood, 2005; Ferreira et al., 2008; Flood, 2010).

P. 7 line 33: Since the supersaturation controls the growth rate but could not be determined  directly with high accuracy in our experiments

P. 8 line 4: The slow growth experiments were done  closer to saturated conditions (typically about 5% supersaturation wrt. ice).

**3.2 Roughness due to humidity cycles**

P. 8 line 30: Another process that could influence the roughness of ice crystals is the exposure to several depositional growth and sublimation cycles , which can occur in the atmosphere (Nelson, 1998; Korolev et al., 1999; Ulanowski et al., 2014).

P. 9 line 28: the cyclic growth described here tended to result in a  reduction in roughness during each sublimation phase.

P. 9 line 29: The apparent disparity between our observations and SEM experiments can be accounted for by the fact that growth in the absence of air that takes place in a SEM chamber, instead of being  limited by vapour diffusion as is the case for ice at tropospheric pressures, becomes limited by the attachment kinetics . This distinction is known to lead to different growth rates as well as habits (Beckmann, 1982; Kuroda and Gonda, 1984; Libbrecht, 2017). Consequently, during SEM observations water molecule removal can take place anywhere on facet surfaces, leading to pronounced roughness. Moreover, the bunchingof elementary molecular steps, possibly due to the Schwoebel effect (Misbah et al., 2010), can result in the creation of larger, microscopic (as opposed to elementary) steps that can be seen in SEM micrographs (Cross, 1969).

P. 10 line 3: A further difference between the diffusion limited and kinetics-limited growth is that the former can lead to increased numbers of faults (Beckmann, 1982) and dendritic, skeletal, or needle-shaped crystals, while the latter tends to produce more perfect, smooth, isometric prisms (Gonda, 1976, 1977)

**3.3 Atmospheric implications**

P. 11 line 2: It is very likely, as shown in our experiments, that at higher supersaturation rougher crystals  tend to develop instead of smoother ones

*Author contributions*: C.C, J.V., P.H., H.B. designed and carried out the experiments including sample preparation and data analysis with assistance from T.C. J.V. performed the computational fluid dynamics simulations. Z.U. conceived and supervised the project and provided crystal property measurement techniques and interpretation of ice growth processes. F.S., Z.U, J.V. developed the main conceptual ideas and the technical details of the experiments and the

experimental setup IRIS. C.C., J.V., Z.U. wrote the manuscript with support from P.H., H.B., F.S, T.C., D.N., S.H. and G.R. All authors discussed the results.

*Acknowledgements:* This work was supported by the UK Natural Environment Research Council grant NE/I020067/1 (ACID-PRUF) and the EU Eurochamp-2 scheme grant E2-2011-12-06-0065. The concept of LISA was proposed by Alexei Kiselev, and the instrument itself was designed and built by Edwin Hirst at the University of Hertfordshire.

**Figures**

Figure 4. Comparison between measured and calculated temperature in dependence of the total flow rate using three different wall temperatures. The solid lines represent interpolation of the experimental data.

Figure 6. 2-D scattering  patterns of a smooth column (left panel) and a rough column (right panel). The patterns were produced using the SID3 instrument in the AIDA cloud chamber during growth at low (left) and high (right) supersaturation (Schnaiter et al., 2016).

**Supplements S3, S4 and S5** (PLEASE NOTE: Figures S3 and S4 are new, Figure S5 was previously Figure 5 in the main text)

[Figure]

Fig. S3. Measured and simulated velocity at the center outlet of the laminar flow diffusion chamber with respect to the total flow. The measurements were done by means of an hot-wire anemometer (Dantec Dynamics, Denmark). The miniature hot-wire sensor (Dantec 55P11 sensor) was fixed at the tube outlet in the measuring volume of the laser beam. Single point measurements were done in dependence of the total flow. The data were compared to Fluent simulation results. Both measurements and simulations were done at 20º and dry conditions.

[Figure]

Fig. S4. Illustration of calculated velocity data (contour plot of Fluent simulation results) and the difference between measurements and calculations (values). The position of the numbers corresponds to the measurement position of the hot-wire sensor in the cross sectional profile at the tube outlet.

 Fig. S5. Contour plot of the temperature characterization results. The figure was obtained by a linear interpolation of the temperature measurements (black dots). The black lines are temperature isolines. The figure illustrates that IRIS can be used over a broad temperature range.

[revised manuscript text omitted]

Figure S3: Measured and simulated velocity at the center outlet of the laminar flow diffusion chamber with respect to the total flow. The measurements were done by means of an hot-wire anemometer (Dantec Dynamics, Denmark). The miniature hot-wire sensor (Dantec 55P11 sensor) was fixed at the tube outlet in the measuring volume of the laser beam. Single point measurements were done in dependence of the total flow. The data were compared to Fluent simulation results. Both, measurements and simulations were done at 20° and dry conditions.

[Figure]

Figure S4: Illustration of calculated velocity data (contour plot of Fluent simulation results) and the difference between measurements and calculations (values). The position of the numbers corresponds to the measurement position of the hot-wire sensor in the cross sectional profile at the tube outlet.

[Figure]

Figure S5: Contour plot of the temperature characterization results. The figure was obtained by a linear interpolation of the temperature measurements (black dots). the black lines are temperature isolines. The figure illustrates that IRIS can be used over a broad temperature range.

---

## Author Response (AR2)

**Responses to Referee #1**
**"Surface roughness during depositional growth and sublimation of ice crystals" *by Cèdric Chou et al.**

Most of the itemized concerns made in my earlier review have been adequately addressed in the revised manuscript. A remaining sticky point concerns speculative mechanistic explanations provided by the authors. The authors statement (at the beginning of section 3.1) that

"Regular, smooth crystals can be grown at low supersaturation, where the growth rate is slow enough for the deposited molecules to diffuse on facets to well-separated attachment sites at steps, kinks, and ledges."

remains problematic. As I pointed out earlier, this statement is not substantiated by any experimental evidence given in the paper. It is also inconsistent with much recent research, for example this statement from PNAS (https://www.ncbi.nlm.nih.gov/pmc/articles/PMC5240720/):

"the picture that emerges is that at 200 K disorder sets in within the top bilayer of ice  Then at 257 K the second bilayer melts and subsequently surface melting proceeds from 257 K onward."

This implies that, in the temperature range relevant to the experiments presented in the ms, one would expect a QLL thickness somewhere between one and two bilayers. The authorss dismissal of this concern (in their response to reviewers) is unconvincing; they claim that the QLL becomes "very low", but the reference they cite (Conde et al, 2008) estimates QLL thicknesses on the order of 3-5 Angstroms at 230 K, i.e., in excess of one bilayer. One monolayer is (it would seem to me) easily enough to dominate ice/vapor interactions. Moreover, other workers (e.g., Kong, 2014, "Molecular investigations ", Neshyba et al 2016, "A quasi-liquid mediated ", and others) have argued that ice/QLL/vapor interactions dominate the conversion of QLL to ice over a broad range of temperatures, and identify a key role to be played by the QLL in growth inhibition and supersaturation thresholds. Why are these findings relevant? Because they argue that the two primary events controlling growth of ice from vapor deposition (capture and conversion) are controlled by the QLL, at least at some temperatures relevant to the atmosphere.

In summary: The description of the surface of ice as a place where deposited molecules diffuse on facets to well-separated attachment sites at steps, kinks, and ledges is not consistent with the consensus view of the ice/vapor interface, and is not supported by experiments presented in the ms. One remedy is to simply remove it, since (as far as I can tell) the conclusions presented by the authors do not depend on this picture of the surface of ice. Another remedy is to to acknowledge research that points to the QLL as playing an important role. I believe the paper will have a longer citation half-life if this is done.

We are grateful to the Reviewer for a detailed clarification of one of the viewpoints presented in the earlier review. However, we would like to resist the temptation to remove the sentence in question regarding the conventional, "layer" view of the growth process, for several reasons. We contend that there is as yet no consensus regarding the quantitative or indeed even qualitative importance of the quasi-liquid layer (QLL), especially at lower temperatures, and certainly in terms of surface roughness. As we said in our earlier response, there is strong evidence going back many decades (e.g. Mason et al., 1963; Sazaki et al., 2010) for the view we have presented. In contrast, much of the work concerning the QLL is recent, so not widely cross-tested, and often contradictory. Much of it is based on molecular dynamics (MD) modelling, with different models showing different outcomes, and on the basis of the atmospherically irrelevant assumption of growth in vacuum (as critically discussed in our paper). The results vary widely. For example, the commentary cited by the Reviewer (Michaelides and Slater, 2017) shows QLL thickness at -30 degC of near 0 from measurement and about 0.3 nm from MD; Sànchez et al. (2017), while finding indications that a thin QLL might be present at 235 K, state: "Despite the general agreement on the presence of a QLL below the bulk freezing point, the temperature-dependent thickness of the QLL has remained controversial. The experimentally reported onset temperature for QLL formation varies between 200 K and 271 K".
However, we suggest the following modification of the text, as a compromise highlighting the possible importance of the QLL and expanding on the role of growth conditions. The earlier text:

> "Regular, smooth crystals can be grown at low supersaturation, where the growth rate is slow enough for the deposited molecules to diffuse on facets to well-separated attachment sites at steps, kinks, and ledges. In contrast, fast growth promotes attach-

> ment anywhere on crystal surface, resulting in roughness. Fast growth can moreover lead to the creation of defects and ionization, which further promote irregular growth (Beckmann, 1982; Dash et al., 2001; Dash and Wettlaufer, 2003; Pantaraks and Flood, 2005; Ferreira et al., 2008; Flood, 2010)."

now becomes:

> "The conventional view of ice (and other material) growth suggests that regular, smooth crystals grow at low supersaturation, where the growth rate is slow enough for the deposited molecules to diffuse laterally on facets to well-separated attachment sites at steps, kinks, and ledges. In contrast, fast growth promotes attachment anywhere on crystal surface through 2-D nucleation, and potentially also step bunching, resulting in roughness (Mason et al., 1963; Beckmann, 1982; Dash and Wettlaufer, 2003; Pantaraks and Flood, 2005; Dash et al., 2006; Ferreira et al., 2008; Flood, 2010; Sazaki et al., 2010). Fast growth can moreover lead to the creation of defects and ionization, which further promote irregular growth (Dash et al., 2001; Dash and Wettlaufer, 2003). Furthermore, the mechanisms leading to surface roughness are likely to depend on growth temperature, in the same way that the gross crystal habit does due to different growth rates on the basal and prismatic facets (Mason et al., 1963; Bailey and Hallett, 2004). Hence roughness may arise on different facets at different temperatures. Also temperature dependent is the role in ice growth of the quasi-liquid layer (QLL). Its thickness, amount of disorder and hence importance diminish with decreasing temperature, with some studies indicating little impact at the temperatures used here but, as yet, there is much disagreement between molecular dynamics modelling and measurements of QLLs (Dash et al., 2006; Gladich et al., 2015; Michaelides and Slater, 2017)."

A very minor point: The line referred to as "Equation (1)" is not an equation, as there is no "=" sign in it.

We changed the word "equation" to "the expression", as suggested.

Furthermore, we inserted a long dash "after roughness in a general sense".

We have also changed the order of authorship, to reflect contributions since

the discussion version of the paper was submitted.

References for Reply:

Dash, J. G., Rempel, A. W., and Wettlaufer, J. S.: The physics of pre-melted ice and its geophysical consequences, Rev. Mod. Phys., 78, 695-741, doi:10.1103/RevModPhys.78.695, 2006.

Gladich, I., Oswald, A., Bowens, N., Naatz, S., Rowe, P., Roeselova, M., and Neshyba, S.: Mechanism of anisotropic surface self-diffusivity at the prismatic icevapor interface, Phys. Chem. Chem. Phys., 17, 22947-22958, doi:10.1039/c5cp01330e, 2015.

Mason, B. J., Bryant G. W., and Van den Heuvel, A. P.: The growth habits and surface structure of ice crystals, Philos. Mag. 8, 505526, doi:10.1080/14786436308211150, 1963.

Michaelides, A., and Slater, B.: Melting the ice one layer at a time, Proc. Natl. Acad Sci., 114, 195197, doi:10.1073/pnas.1619259114, 2017.

Sànchez, M.A., Kling, T., Ishiyama, T., van Zadel, M.J., Bisson, P.J., Mezger, M., Jochum, M.N., Cyran, J.D., Smit, W.J., Bakker, H.J., and Shultz, M.J.: Experimental and theoretical evidence for bilayer-by-bilayer surface melting of crystalline ice, Proc. Natl. Acad Sci., 114, 227-232, doi:10.1073/pnas.1612893114, 2017.

[revised manuscript text omitted]